# Knowledge Distillation:
# Bad Models Can Be Good Role Models

**Gal Kaplun**
Harvard University & Mobileye
galkaplun@g.harvard.edu

**Eran Malach**
Hebrew University & Mobileye
eran.malach@mail.huji.ac.il

**Preetum Nakkiran**
University of California San Diego
preetum@ucsd.edu

**Shai Shalev-Shwartz**
Hebrew University & Mobileye
shais@cs.huji.ac.il

## Abstract

Large neural networks trained in the overparameterized regime are able to fit noise to zero train error. Recent work of Nakkiran and Bansal [20] has empirically observed that such networks behave as "conditional samplers" from the noisy distribution. That is, they replicate the noise in the train data to unseen examples. We give a theoretical framework for studying this conditional sampling behavior in the context of learning theory. We relate the notion of such samplers to knowledge distillation, where a student network imitates the outputs of a teacher on unlabeled data. We show that samplers, while being bad classifiers, can be good teachers. Concretely, we prove that distillation from samplers is guaranteed to produce a student which approximates the Bayes optimal classifier. Finally, we show that some common learning algorithms (e.g., Nearest-Neighbours and Kernel Machines) can often generate samplers when applied in the overparameterized regime.

## 1   Introduction

Recently, the field of supervised learning has witnessed the success of *overparameterized* methods: models, such as deep neural networks, which are large enough to fit their train sets but still achieve good test performance. A core theoretical concern is to understand why such models are able to fit even noisy training data without catastrophically overfitting [31] despite no explicit regularization. The seminal work of Bartlett et al. [2] proposed the theoretical framework of *benign overfitting* to capture this empirical behavior. Briefly, benign overfitting studies *statistically consistent* methods— where models approach the Bayes optimal classifier, even in presence of noise.

However, recent empirical work shows that when training deep neural networks on noisy data, *overfitting is neither catastrophic nor benign* [20]. Specifically, they propose that overfitting leads not to a good classifier, but to a good *conditional sampler*. For example, suppose we train a model on a set of images sampled from some distribution $\mathcal{D}$, where $20\%$ of the images of cats are wrongly labeled as dogs. We now train an overparameterized network to fit samples from $\mathcal{D}$. Note that for the distribution $\mathcal{D}$, the Bayes-optimal classifier, namely $f_{\mathcal{D}}^*(\mathbf{x}) := \operatorname{argmax}_y \mathbb{P}_{\mathcal{D}}(y|\mathbf{x})$, returns the "correct" class of every image. We can hope that if overfitting is truly "benign" in the sense of [2], the overparameterized model will be close to this optimal $f_{\mathcal{D}}^*$. However, this is not what occurs in practice: as [20] point out, the trained model $f$ reproduces noise in the training set at test time, labeling up to $20\%$ of the cats in the *test* data as dogs. In a sense, the trained model $f$ behaves as a conditional sample: $f(\mathbf{x}) \sim \mathbb{P}_{\mathcal{D}}(y|\mathbf{x})$ (see the leftmost confusion matrix in Figure 1).

The above example indicates that thinking about classifiers in the overparameterized regime as approximating the Bayes optimal predictor might be misleading. Therefore, it is essential to develop

36th Conference on Neural Information Processing Systems (NeurIPS 2022).

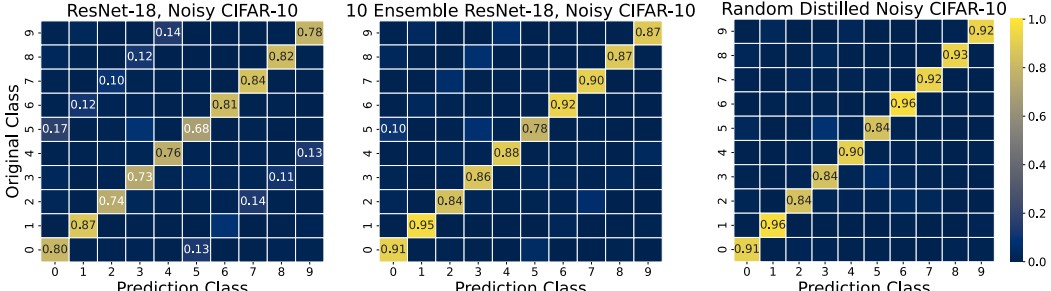

Figure 1: **From Samplers to Good Models.** Class confusion matrices for ResNet-18 trained on CIFAR-10 with 20% fixed and class dependent label noise (e.g., 20% of cats are labeled as dogs). *Left.* A ResNet-18 trained on this data *replicates* the noise to the test distribution. *Middle.* When ensembling 10 such models, the noise virtually disappears, at the cost of high price at inference. *Right.* Distilling a single model via unlabeled examples from the CIFAR-5m dataset by randomly selected a teacher for each example from a 10 teacher pool eliminates the noise as well as the inference cost. For further details on random teacher distillation see Section 3.3

the appropriate theoretical framework for describing the behavior of *samplers* from the conditional distribution. While the learning-theoretic aspects of supervised classification are well-studied, the theory of supervised *conditional sampling* has, to the best of our knowledge, not been systematically explored. In this work, we take the first steps towards addressing this gap. We initiate the study of conditional sampling as a learning problem, and explore its relations to other kinds of learning.

Interestingly, we relate the notion of samplers to knowledge distillation methods. In short, knowledge distillation is the process of training a teacher network on a small labeled dataset and using its predictions to label a large unlabeled dataset, on which a student network is trained to imitate the output of the teacher. We find that taking an ensemble of samplers as a teacher for knowledge distillation produces a student network with minimal error with respect to the Bayes optimal classifier. Finally, our theory leads to a new algorithm for knowledge distillation, where we randomly choose a teacher from a fixed pool, to label each example, which accelerates the training process in practice[1]. We show that this new algorithm is guaranteed to find a student with low error.

## 1.1 Our Contributions

**Learning Theory of Conditional Samplers.** (**Section 2**) We initiate the study of *conditional sampling* in the context of computational learning theory. We formally introduce the problem of conditional-sampling, and define the sample complexity of learning a sampler. We then present positive and negative results in this new setting. For example, we show that there exist distributions where *sampling* is much easier than *classification*, requiring far fewer samples. However, if we allow polynomial blowup in sample-size and runtime, a sampler can be "boosted" to a good classifier, showing that if finding a classifier is computationally hard then sampling is also hard.

**Theory of Knowledge Distillation. (Section 3)** One way to boost a sampler into a good classifier is to run an ensemble of samplers at inference time (see middle panel of Figure 1), which is costly. We show that by performing distillation from an ensemble of teachers, it is possible to find a student with low error w.r.t. the Bayes optimal. This shows that teacher networks in ensemble-distillation need not be good classifiers, but just good samplers. Finally we propose a new algorithm for distillation, where each example is labeled by a random teacher from a fixed pool (see right panel of Figure 1). We study quantitative bounds on the sample complexity of teaching and learning in our setting, for both ensemble-distillation and distillation from a random teacher.

**Theory of Sampler Algorithms. (Section 4)** We show that several classical learning algorithms provably produce good conditional samplers, and analyze their sample complexity in terms of standard problem parameters (e.g., distributional smoothness). Specifically we show that the 1-Nearest-Neighbour algorithm is a sampler, and extend our analysis to k-Nearest-Neighbour as well.

---

[1]See repository `https://github.com/GalKaplun/sampler-distillation`

Furthermore, we show that under some distributional assumptions, Lipschitz classes such as linear methods, kernels and neural networks may also behave like samplers.

## 1.2 Related Work

**Knowledge Distillation** Knowledge Distillation for deep learning was proposed in [14]. Since then, a large body of work showed its practical benefits for various machine learning tasks [29, 23, 30, 6, 10, 27]. Nonetheless, from a theoretical perspective, understanding why and when distillation works remains a mystery. Hinton et al. [14], attribute the success of distillation to the fact that the soft labels of the teacher passing additional information on the input. In a recent paper, Menon et al. [18], claim that when the teacher approximates the Bayes class-probabilities, distillation is possible. Our results show that even when the teachers are far from the Bayes optimal prediction (i.e., when teachers are noisy samplers from the distribution), we can find a student with low levels of noise through distillation. Another work by Wei et al. [28], shows that, assuming the data distribution has good continuity within each class, self-distillation is possible. However, it is not clear when such assumption is actually satisfied. The work of Lopez-Paz et al. [17], relates the notion of privileged information to knowledge distillation. A work by Frei et al. [8], shows that self-training can boost weak learners, when the target is a linear classifier over a mixture model distribution. Other works [19, 32, 7] study distillation as a regularization method, forcing the student to learn under a "smoother" loss landscape.

**Learning with Noise** Learning under corrupted labels is a well-studied research area in the literature (e.g., [1, 9]). A notable examples are the works of Blum et al., and Kearns et al. [5, 15], showing how statistical query algorithms can be leveraged to learn under noisy labels. A growing number of works study the effects of noise on deep learning, as well as methods to learn under label noise (see [26] for a survey). However, most of these works do not leverage distillation for label noise robustness, as we suggest in our work. More related to our work is [12], showing that an iterative process of training and re-labeling can combat label noise. However, this work focuses on empirical study of distillation.

## 2 Sampling as a Learning Problem

Let $\mathcal{X}$ be the input space and $\mathcal{Y} = \{\pm 1\}$ be the label space (for simplicity, under appropriate assumptions we can extend most of our results to multiclass). A hypothesis class $\mathcal{H}$ is some class of functions from $\mathcal{X}$ to $\mathcal{Y}$. A learning algorithm $\mathcal{A}$ takes a sequence of $m$ samples $S \in (\mathcal{X} \times \mathcal{Y})^m$ and outputs some hypothesis $h : \mathcal{X} \to \mathcal{Y}$. We denote by $\mathcal{A}(S)$ the hypothesis that $\mathcal{A}$ outputs when observing the sample $S$. For some distribution $\mathcal{D}$ over $\mathcal{X} \times \mathcal{Y}$, we denote its $\mathcal{X}$ marginal as $\mathcal{D}_\mathcal{X}$ and denote the Bayes optimal classifier by $f_\mathcal{D}^*(\mathbf{x}) := \arg\max_{y \in \mathcal{Y}} \mathbb{P}_\mathcal{D}[y|\mathbf{x}]$.[2]

When $\mathcal{D}$ is a distribution where the label is not a deterministic function of the input, we think of $\mathcal{D}$ as a noisy version of some clean distribution $\mathcal{D}^*$, where each input is correctly labeled. Naturally, we assume that the probability of seeing the right label is greater than seeing a wrong one. In other words, $\mathcal{D}^*$ has the same marginal distribution over $\mathcal{X}$ (i.e., $\mathcal{D}_\mathcal{X} = \mathcal{D}_\mathcal{X}^*$), and is labeled by the Bayes optimal classifier of $\mathcal{D}$. That is, sampling $(\mathbf{x}, y) \sim \mathcal{D}^*$ is given by $\mathbf{x} \sim \mathcal{D}_\mathcal{X}$ and $y = f_\mathcal{D}^*(\mathbf{x})$.

We denote the "noise" of the distribution $\mathcal{D}$ by $\eta(\mathcal{D}) := \mathbb{P}_{(\mathbf{x},y) \sim \mathcal{D}}[y \neq f_\mathcal{D}^*(\mathbf{x})]$. We also consider the margin of the distribution, which defines the difference in probability between correct and wrong label for each example. Namely, for some $\delta \geq 0$, let $\gamma_\delta(\mathcal{D})$ be the supremum over $\gamma > 0$ s.t.,

$$\mathbb{P}_{\mathbf{x} \sim \mathcal{D}_\mathcal{X}} \left[ \mathbb{P}_\mathcal{D}(f_\mathcal{D}^*(\mathbf{x})|\mathbf{x}) < \max_{y \neq f_\mathcal{D}^*(\mathbf{x})} \mathbb{P}_\mathcal{D}(y|\mathbf{x}) + \gamma \right] \leq \delta$$

Specifically, we denote $\gamma(\mathcal{D}) := \gamma_0(\mathcal{D})$. We typically assume that the input distribution has a strictly positive margin $\gamma(\mathcal{D}) > 0$, so for every example the probability to see the correct label is greater by $\gamma$ than the probability to see a wrong label.

Our objective in this setting is to approximate the Bayes optimal classifier of $\mathcal{D}$. In other words, we want to minimize the 0-1 loss on the clean distribution $\mathcal{D}^*$:

$$L_{\mathcal{D}^*}(h) := \mathbb{P}_{(\mathbf{x},y) \sim \mathcal{D}}[h(\mathbf{x}) \neq f_\mathcal{D}^*(\mathbf{x})] = \mathbb{P}_{(\mathbf{x},y) \sim \mathcal{D}^*}[h(\mathbf{x}) \neq y] \tag{1}$$

So, the learning algorithm has access to samples from the noisy distribution $\mathcal{D}$, but needs to achieve good loss on the clean distribution $\mathcal{D}^*$. We define a *learner* in this setting to be an algorithm that minimizes (1) using a finite number of samples:

---

[2]We assume that ties are broken arbitrarily

**Definition 1.** *For some learning algorithm $\mathcal{A}$ and some distribution $\mathcal{D}$ over $\mathcal{X} \times \mathcal{Y}$, we say that $\mathcal{A}$ is a **learner** for $\mathcal{D}$ if there exists a function $m : (0,1) \to \mathbb{N}$ s.t. for every $\epsilon \in (0,1)$, taking $m \geq m(\epsilon)$ we get, $\mathbb{E}_{S \sim \mathcal{D}^m} L_{\mathcal{D}^*}(\mathcal{A}(S)) \leq \epsilon$.*

*In this case, we call $m(\cdot)$ the sample complexity of $\mathcal{A}$ w.r.t. $\mathcal{D}$. Additionally, for some class $\mathcal{P}$ of distributions over $\mathcal{X} \times \mathcal{Y}$, we say that $\mathcal{A}$ is a **learner** for $\mathcal{P}$ if there is some $m(\cdot)$ s.t. $\mathcal{A}$ is a learner with sample complexity $m(\cdot)$ for every $\mathcal{D} \in \mathcal{P}$.*

Note that this definition of learner is similar to the notion of asymptotically consistent estimator. However, our definition explicitly accounts for the sample complexity. To give a concrete example, let us consider agnostic learning using the ERM rule, namely: $\mathrm{ERM}_{\mathcal{H}}(S) = \arg\min_{h \in \mathcal{H}} L_S(h)$.

For some hypothesis class $\mathcal{H}$ and some margin $\gamma > 0$, let $\mathcal{P}(\mathcal{H}, \gamma)$ be the class of distributions such that for every $\mathcal{D} \in \mathcal{P}(\mathcal{H}, \gamma)$ we have $\gamma(\mathcal{D}) \geq \gamma$ and $f_{\mathcal{D}}^* \in \mathcal{H}$. Namely, $\mathcal{P}(\mathcal{H}, \gamma)$ is the class of distributions with margin $\gamma$ for which the Bayes optimal classifier comes from $\mathcal{H}$. The following theorem states that finite VC-dimension and a non-zero margin form a sufficient condition for learnability with noise.

**Theorem 2.** *There exists a constant $C > 0$ s.t. for every hypothesis class $\mathcal{H}$ with $\mathrm{VC}(\mathcal{H}) < \infty$, $\mathrm{ERM}_{\mathcal{H}}$ is a **learner** for $\mathcal{P}(\mathcal{H}, \gamma)$ with sample complexity $m(\epsilon) = C \frac{\mathrm{VC}(\mathcal{H}) + \log(1/\epsilon)}{\epsilon^2 \gamma^2}$.*

The proof of Theorem 2 is given in the appendix. The main idea is the following lemma (whose proof is also in the appendix), which shows that the margin assumption connects a small relative error w.r.t. $\mathcal{D}$ to a small absolute error w.r.t. $\mathcal{D}^*$.

**Lemma 3.** *Fix some distribution $\mathcal{D}$, and let $h^*$ be the Bayes optimal classifier for $\mathcal{D}$. Assume that $\gamma_\delta(\mathcal{D}) > 0$. Then, for every $h$ such that $L_{\mathcal{D}}(h) \leq L_{\mathcal{D}}(h^*) + \epsilon$ it holds that $L_{\mathcal{D}^*}(h) \leq \frac{\epsilon}{\gamma_\delta(\mathcal{D})} + \delta$.*

Combining this lemma with the Fundamental Theorem of Learning Theory (see [25]) yields Theorem 2. From this theorem, we see that given more samples than the VC-dimension (often corresponding to the number of parameters), learning is possible. However, complex classifiers with large VC-dimension such as neural networks are often trained in the *overparameterized* regime, when the number of parameters exceeds the number of samples. In this regime, the bound of Theorem 2 becomes vacuous. That said, this does not rule out the possibility of distribution-dependent sample complexity bounds using other measures of complexity (e.g., Rademacher complexity).

## 2.1 Samplers

As previously mentioned, neural networks trained on noisy data replicate the noise to unseen samples as well, behaving like *samplers* from the noisy distribution. We next define formally a sampler for some distribution $\mathcal{D}$, giving a similar definition as in [20].

For some learning algorithm $\mathcal{A}$, some number $m \in \mathbb{N}$ and some distribution $\mathcal{D}$ over $\mathcal{X} \times \mathcal{Y}$, define the distribution $\mathcal{A}(\mathcal{D}^m)$ over $\mathcal{X} \times \mathcal{Y}$, where $(\mathbf{x}, y) \sim \mathcal{A}(\mathcal{D}^m)$ is given by sampling $S \sim \mathcal{D}^m$, sampling $\mathbf{x} \sim \mathcal{D}_{\mathcal{X}}$ and setting $y = \mathcal{A}(S)(\mathbf{x})$. Namely, $\mathcal{A}(\mathcal{D}^m)$ is the distribution given by (re)-labeling $\mathcal{D}$ using a hypothesis generated by $\mathcal{A}$ when observing a random sample of size $m$. Using these notations, we define a sampler algorithm for the distribution $\mathcal{D}$ as follows:

**Definition 4.** *For some learning algorithm $\mathcal{A}$ and some distribution $\mathcal{D}$ over $\mathcal{X} \times \mathcal{Y}$, we say that $\mathcal{A}$ is a **sampler** for $\mathcal{D}$ if there exists $\widetilde{m} : (0,1) \to \mathbb{N}$ s.t. for every $\epsilon \in (0,1)$, taking $m \geq \widetilde{m}(\epsilon)$ we get,*

$$\mathrm{TV}(\mathcal{A}(\mathcal{D}^m), \mathcal{D}) \leq \epsilon$$

*where $\mathrm{TV}$ is the Total Variation Distance. Then, we call $\widetilde{m}$ the sample complexity of $\mathcal{A}$ w.r.t. $\mathcal{D}$. Additionally, for some class $\mathcal{P}$ of distributions over $\mathcal{X} \times \mathcal{Y}$, we say that $\mathcal{A}$ is a **sampler** for $\mathcal{P}$ if there exists $\widetilde{m}(\cdot)$ s.t. $\mathcal{A}$ is a sampler with sample complexity $\widetilde{m}(\cdot)$ for each $\mathcal{D} \in \mathcal{P}$.*

So, a sampler for $\mathcal{D}$ is an algorithm that generates a distribution similar to $\mathcal{D}$ (the noisy input distribution) when labeling new examples. A primary example for a sampler is the 1-Nearest-Neighbour algorithm: since 1-NN outputs the (possibly corrupted) label of the closest neighbour, its prediction behaves like sampling from $\mathbb{P}_{\mathcal{D}}(y|\mathbf{x})$ (see Section 4). Figure 1 shows that neural networks behave similarly to samplers when trained on a noisy version of the CIFAR-10 dataset.

Given the above definition, it can be easily shown that, instead of approximating the Bayes optimal prediction, a sampler preserves the noise rate of the original distribution.

**Lemma 5.** *Let $\mathcal{A}$ be a **sampler** $\mathcal{D}$ with sample complexity $\tilde{m}$. Then, for $m \geq \tilde{m}(\varepsilon)$,*

$$\eta(\mathcal{D}) - \varepsilon \leq \mathop{\mathbb{E}}_{S \sim \mathcal{D}^m} L_{\mathcal{D}^*}(\mathcal{A}(S)) \leq \eta(\mathcal{D}) + \varepsilon$$

While a *sampler* for $\mathcal{D}$ is not a good *learner* (in the sense of Definition 1), it does have some favorable properties. Primarily, it can take significantly fewer samples to get a sampler than it would take to get a learner, hence making the study of samplers more suitable for the overparameterized regime. In fact, in the extreme case one could get a sampler using only a *single example* from the distribution, while getting a learner for the same distribution would require an arbitrarily large number of examples. Indeed, fix $b \in \{\pm 1\}$ and $\gamma \in (0,1)$, and let $\mathcal{D}_b$ be the distributions concentrated on a single example $\mathbf{x} \in \mathcal{X}$, with label $y \in \{\pm 1\}$ s.t. $\mathbb{P}_{\mathcal{D}_b}(y = 1) = \frac{1+b\gamma}{2}$. To get a sampler from $\mathcal{D}_b$ it clearly suffices to take a single example $(\mathbf{x}, y)$, and return the constant function $y$. To find the Bayes optimal for the distribution $\mathcal{D}_b$, on the other hand, any algorithm needs $\Omega(1/\gamma^2)$ examples. Using this observation, we show the following result:

**Theorem 6.** *For every $M > 0$, there exists a class of distributions $\mathcal{P}_M$ such that: a) there exists a **sampler** for $\mathcal{P}_M$ with sample complexity $\widetilde{m} \equiv 1$, and b) any **learner** for $\mathcal{P}_M$ has sample complexity satisfying $m(1/8) \geq M$.*

## 2.2  Teachers

Motivated by the observed behavior of neural networks in the overparemeterized regime, we defined the notion of *samplers*. We showed that samplers can be much more sample efficient than learners, at the cost of giving noisy predictions. Next, we will show that while samplers are in and of themselves bad classifiers, they can still be good *teachers*. That is, we can use samplers to label a large unlabeled dataset and train a *student* classifier on this new dataset. This process is often referred to as *knowledge distillation*, and it has been shown to work remarkably well in practice [29, 23]. The following definition captures the notion of a (good) teacher:

**Definition 7.** *For some learning algorithm $\mathcal{A}$, some distribution $\mathcal{D}$ over $\mathcal{X} \times \mathcal{Y}$, we say that $\mathcal{A}$ is a **teacher** for $\mathcal{D}$ if there exists a function $\widetilde{m} : (0,1)^2 \to \mathbb{N}$ s.t. for every $\epsilon, \tau \in (0,1)$, taking $m \geq \widetilde{m}(\epsilon, \tau)$ the following holds:*

*1. $L_{\mathcal{D}^*}\left(f^*_{\mathcal{A}(\mathcal{D}^m)}\right) = \mathbb{P}_{\mathbf{x} \sim \mathcal{D}}\left[f^*_{\mathcal{D}}(\mathbf{x}) \neq f^*_{\mathcal{A}(\mathcal{D}^m)}(\mathbf{x})\right] \leq \epsilon$*

*2. $\gamma_\epsilon(\mathcal{A}(\mathcal{D}^m)) \geq \gamma(\mathcal{D}) - \tau$*

*Additionally, for some class $\mathcal{P}$ of distributions over $\mathcal{X} \times \mathcal{Y}$, we say that $\mathcal{A}$ is a **teacher** for $\mathcal{P}$ if there is some $\widetilde{m}(\cdot, \cdot)$ s.t. $\mathcal{A}$ is a teacher with sample complexity $m(\cdot, \cdot)$ for every $\mathcal{D} \in \mathcal{P}$.*

The first condition in the above definition means that the Bayes optimal of the original distribution and the distribution induced by the teacher are close. The second condition means that the probability mass of low margin samples from the distribution labeled by the teacher is small. Intuitively, an algorithm satisfying Definition 7 is a good teacher since the distribution it induces is "similar" to the original distribution. Hence, if the student finds a good hypothesis on the teacher-induced distribution, its hypothesis is also good w.r.t. the original distribution.

In Section 3 we formally study when and how teachers can be used for distillation, giving guarantees for getting students with small error with respect to the clean distribution $\mathcal{D}^*$. Before that, let us first show that samplers are indeed good teachers.

**Theorem 8.** *Let $\mathcal{A}$ be a **sampler** for $\mathcal{D}$ with sample complexity $m(\cdot)$ and margin $\gamma$. Then, $\mathcal{A}$ is a **teacher** for $\mathcal{D}$ with sample complexity $\widetilde{m}(\epsilon, \tau) = m(\varepsilon \cdot \min(\tau/2, \gamma))$.*

The main idea behind the proof of Theorem 8 is that, given an example with probability mass $p$, the "cost" (in terms of TV) of flipping the example's label w.r.t. the Bayes optimal $f^*_{\mathcal{D}}$ is at least $p \cdot \gamma$. Since our TV "budget" is limited by $\varepsilon\gamma$ we can only flip a probability mass of $\varepsilon$ of the distribution.

Finally, before moving on to discuss the implication of our results, we show that learners are also good teachers. This is almost immediate, since learners approximate the Bayes optimal classifier, and hence can be used to train students to similarly imitate the Bayes classifier.

**Theorem 9.** *Let $\mathcal{A}$ be a **learner** for $\mathcal{D}$ with sample complexity $m(\cdot)$. Then, $\mathcal{A}$ is a **teacher** for $\mathcal{D}$ with sample complexity $\widetilde{m}(\epsilon, \tau) = m\left(\frac{\epsilon(1 - \gamma(\mathcal{D}) + \tau)}{2}\right)$.*

To conclude, Theorem 8 and Theorem 9 show that, to some extent, a teacher is an "interpolation" between a sampler and a learner.

# 3 Distillation from Teachers

We defined the notion of a teacher, and showed that both samplers and learners are teachers. Now, we show how teachers (and in particular, samplers) can be used to find good learners. First, we show that an ensemble of teachers can be used to get a good learner by simply outputting the majority vote of the ensemble. Next, we show that using the ensemble to label a new set of unlabeled examples (i.e., performing knowledge distillation) guarantees finding a student with small loss, assuming the Bayes optimal classifier comes from the hypothesis class learned by the student. Such process is favorable, since it reduces the computational cost of running the ensemble at inference time, and also allows using a different hypothesis class for the student (for example when using a student network of smaller size, e.g., [11]). Finally, we show that distillation can also be achieved by labeling examples using a teacher that is randomly chosen from a fixed set of teachers, a method that has some computational benefits at training time. To the best of our knowledge, this is a novel technique for distillation that has not been previously suggested in prior work.

## 3.1 Ensembles of Teachers

We now show how an ensemble of teachers can be used to get accurate predictions with respect to the Bayes optimal predictor. Given some $k$ samples $S_1, \ldots, S_k \sim \mathcal{D}^m$, each one of size $m$, we can use a learning algorithm $\mathcal{A}$ to get $k$ different hypothesis $h_1, \ldots, h_k$, where $h_i := \mathcal{A}(S_i)$. Observe the ensemble hypothesis, which outputs the majority vote of the ensemble members:

$$h_{\text{ens}}(\mathbf{x}) = \arg\max_{y \in \mathcal{Y}} \sum_i \mathbf{1}\{h_i(\mathbf{x}) = y\}$$

We use the notation $\mathcal{A}_{\text{ens}}(S_1, \ldots, S_k) := h_{\text{ens}}$ to denote this ensemble hypotheses. The following Theorem states that the ensemble hypothesis has a good loss on average, when using a large enough ensemble of teachers:

**Theorem 10.** *Assume that $\mathcal{A}$ is a **teacher** for some distribution $\mathcal{D}$ with complexity $\widetilde{m}$. Then, for all $\epsilon \in (0, 1)$, taking $m \geq \widetilde{m}\left(\frac{\epsilon}{3}, \frac{\gamma(\mathcal{D})}{2}\right)$ and $k \geq \frac{16 \log(3/\epsilon)}{\gamma(\mathcal{D})^2}$ we get,*

$$\mathbb{E}_{S_1, \ldots, S_k \sim \mathcal{D}^m} L_{\mathcal{D}^*}(\mathcal{A}_{\text{ens}}(S_1, \ldots, S_k)) \leq \epsilon$$

We now give a sketch of the proof. For some $\mathbf{x} \in \mathcal{X}$, let $y_i$ be the prediction of the $i$-th teacher, and let $\bar{y}$ be the average of the predictions, namely $\bar{y} = \frac{1}{k} \sum_i y_i$. Observe that $h_{\text{ens}}(\mathbf{x}) = \text{sign}(\bar{y})$, and additionally $\mathbb{E}[\bar{y}] = \mathbb{E}_{\mathcal{A}(\mathcal{D}^m)}[y|\mathbf{x}]$. Now, since $\mathcal{A}$ is a teacher we get $\mathbb{E}_{\mathcal{A}(\mathcal{D}^m)}[y|\mathbf{x}] \approx \mathbb{E}_{\mathcal{D}}[y|\mathbf{x}]$ (with high probability over the choice of $\mathbf{x}$), and by concentration bounds this implies that w.h.p. $h_{\text{ens}}(\mathbf{x})$ give the Bayes optimal prediction (i.e., $h_{\text{ens}}(\mathbf{x}) = f_{\mathcal{D}}^*(\mathbf{x})$).

So, Theorem 10 shows that if $\mathcal{A}$ is a **teacher** for $\mathcal{D}$, then $\mathcal{A}_{\text{ens}}$ with $k \geq \widetilde{\Omega}\left(1/\gamma(\mathcal{D})^2\right)$[3] is a **learner** for $\mathcal{D}$. More generally, if we have a teacher for some distribution $\mathcal{D}$ with positive margin, this implies that there exists a learner for the same distribution. The inverse of this statement gives another interesting result—if no algorithm can learn some problem, then getting a teacher (or sampler) is also hard. Formally, let $\mathcal{P}$ be a class of distributions over $\mathcal{X} \times \mathcal{Y}$ s.t. for all $\mathcal{D} \in \mathcal{P}$ it holds that $\gamma(\mathcal{D}) \geq \gamma$. Then, if there is no **learner** for $\mathcal{P}$, there is no **teacher** or **sampler** for $\mathcal{P}$. Additionally, a similar result holds for problems which are computationally hard to learn. That is, if there is no learner for $\mathcal{P}$ that runs in polynomial time, then there is no poly-time teacher or sampler for $\mathcal{P}$. Observe that the condition that $\gamma(\mathcal{D}) \geq \gamma$ for all $\mathcal{D} \in \mathcal{P}$ in the previous statement is necessary. Indeed, taking $\mathcal{P} = \cup_{M=1}^{\infty}$, where $\mathcal{P}_M$ is the distribution class guaranteed by Theorem 6, gives a class $\mathcal{P}$ s.t. there is a sampler for $\mathcal{P}$ with sample complexity $\widetilde{m} \equiv 1$, but there is no learner for $\mathcal{P}$.

## 3.2 Distillation from Ensembles

We showed that an ensemble of $k = \tilde{\Omega}(\gamma^{-2})$ teachers gives a classifier that approximates the Bayes optimal predictor. This, however, incurs a $k$ factor in computational cost at inference time. To prevent

---

[3]We use $\tilde{\Omega}$ to hide constant and logarithmic factors.

this, we can instead use the ensemble to label new unlabeled data, and train a new classifier to imitate the ensemble. This moves the computational burden from inference time to training time.

Fix some class $\mathcal{H}$, and define the **Ensemble-Pseudo-Labeling (EPL)** algorithm as follows:

1. For some $k, m \in \mathbb{N}$, sample $S_1, \ldots, S_k \sim \mathcal{D}^m$.
2. Run $\mathcal{A}$ on $S_1, \ldots, S_k$, and let $h_{\mathrm{ens}} = \mathcal{A}_{\mathrm{ens}}(S_1, \ldots, S_k)$.
3. Take $S'$ to be a set of $m'$ **unlabeled** examples sampled from $\mathcal{D}_{\mathcal{X}}$, and label it using $h_{\mathrm{ens}}$.
4. Denote by $\widetilde{S}$ the pseudo-labeled set. Run $\mathrm{ERM}_{\mathcal{H}}$ on the set $\widetilde{S}$ and return $h := \mathrm{ERM}_{\mathcal{H}}(\widetilde{S})$.

Intuitively, since $h_{\mathrm{ens}}$ approximates the Bayes optimal classifier (see Theorem 10), the labels for the new dataset $S'$ are mostly correct. In other words, the pseudo-labeled set $\widetilde{S}$ comes from a distribution that is close to the clean distribution $\mathcal{D}^*$. When using pseudo-labels, it is enough to use unlabeled data, which is often abundant, so $\widetilde{S}$ can be much larger than our original labeled dataset. In this case, we no longer need to work in the overparameterized regime, so $\mathrm{ERM}_{\mathcal{H}}$ is guaranteed to achieve good performance by standard VC bounds. This argument is captured in Theorem 11:

**Theorem 11.** *For hypothesis class $\mathcal{H}$ with $\mathrm{VC}(\mathcal{H}) < \infty$, and let $\mathcal{D} \in \mathcal{P}(\mathcal{H}, \gamma)$ for some $\gamma > 0$. Let $\mathcal{A}$ be a **teacher** for $\mathcal{D}$ with distributional sample complexity $\widetilde{m}$. Then, there exists a constant $C > 0$ s.t. for every $\epsilon \in (0,1)$, running the **EPL** algorithm with parameters $m \geq \widetilde{m}\left(\frac{\epsilon}{12}, \frac{\gamma}{2}\right)$, $m' \geq C\frac{\mathrm{VC}(\mathcal{H}) + \log(1/\epsilon)}{\epsilon^2}$ and $k \geq \frac{16\log(12/\epsilon)}{\gamma^2}$ returns a hypothesis $h$ satisfying $\mathbb{E}_{S_1, \ldots, S_k, \widetilde{S}} L_{\mathcal{D}^*}(h) \leq \epsilon$.*

### 3.3 Distillation from Random Teachers

While the **EPL** algorithm moves the computation cost from inference to training, we still suffer a $k$ factor for labeling each example. Instead, a possible solution is to label examples by choosing a random classifier from the ensemble. Then, we only run one classifier per example. So, we define the **Random-Pseudo-Labeling (RPL)** similarly to **EPL**, except that for each example $\mathbf{x}$ in the unlabeled dataset $S'$, we pick $h \sim \{h_1, \ldots, h_k\}$ and label $\mathbf{x}$ by $h(\mathbf{x})$.

To understand why this method works, we can think of $\widetilde{S}$ as coming from a distribution $\widetilde{\mathcal{D}}$, defined by sampling $\mathbf{x} \sim \mathcal{D}_{\mathcal{X}}$ and sampling $y$ s.t. $\mathbb{E}_{\widetilde{\mathcal{D}}}[y|\mathbf{x}] = \mathbb{E}_{i\sim[k]}[h_i(\mathbf{x})]$. By the properties of the teacher, using concentration arguments as in Theorem 10, the distribution $\widetilde{\mathcal{D}}$ is close to the *noisy* distribution $\mathcal{D}$. However, as mentioned before, the advantage of using $\widetilde{S} \sim \widetilde{\mathcal{D}}$ is that we can use a much larger set of unlabeled data, in which case the result of Theorem 2 can be applied to show that the above algorithm finds a hypothesis with good error. This is stated in the following result:

**Theorem 12.** *For hypothesis class $\mathcal{H}$ with $\mathrm{VC}(\mathcal{H}) < \infty$. let $\mathcal{D} \in \mathcal{P}(\mathcal{H}, \gamma)$ for $\gamma > 0$. Let $\mathcal{A}$ be a **teacher** for $\mathcal{D}$ with distributional sample complexity $\widetilde{m}$. Then, there exists a constant $C > 0$ s.t. for every $\epsilon \in (0,1)$, running the **RPL** algorithm with parameters $m \geq \widetilde{m}\left(\frac{\epsilon\gamma}{54}, \frac{\gamma}{2}\right)$, $m' \geq C\frac{\mathrm{VC}(\mathcal{H}) + \log\left(\epsilon^{-1}\gamma^{-1}\right)}{\epsilon^2\gamma^2}$ and $k \geq \frac{128\log\left(36\epsilon^{-1}\gamma^{-1}\right)}{\gamma^2}$ returns $h$ satisfying $\mathbb{E}_{S_1, \ldots, S_k, \widetilde{S}} L_{\mathcal{D}^*}(h) \leq \epsilon$.*

Compare the sample complexity of the above theorem with the sample complexity achieved by the EPL algorithm, stated in Theorem 11. At first glance, it seems that the gain from using the RPL algorithm is not clear, as it increases the number of unlabeled data by a factor of $1/\gamma(\mathcal{D})^2$ (and also might increase the number of labeled examples). This is not surprising, because a dataset labeled by a random classifier can be much more noisy than a dataset labeled by the ensemble, and hence more examples are required in order to learn. Note, that since $k \geq \tilde{\Omega}(1/\gamma(\mathcal{D})^2)$, the randomized labeling takes $1/k$ compute per example relative to the ensemble labeling, it will, however, need to label an order of $k$ times more examples.

We do note that there might still be computational benefits to using the random approach, as it allows the training and labeling to happen in parallel without a significant increase in compute (e.g., 2 GPU cores are enough to train in parallel without any loss of time). On the other hand, labeling a dataset using the ensemble requires one to either increase the compute by $k$ (applying parallelism), or otherwise wait until the full dataset is labeled, and only then start the distillation process. Finally, our experiments show that the gain from ensemble labeling over random labeling, as captured by the final accuracy of the trained student, is not so significant (see Table 1). This suggests that in

some particular cases (e.g., under further assumptions on the data distribution and/or the optimization algorithm), it is preferable to use the RPL algorithm.

# 4 Samplers Exist

So far, we showed that samplers can be good teachers. These can then be used for knowledge distillation, generating students which approximate the Bayes optimal prediction. To complete the picture, we now show that some well-known algorithms can be used as samplers or teachers. We start by studying the k-Nearest-Neighbour algorithm, and show that it is a teacher when the underlying distribution has some Lipschitz property. We then show that bounded-norm ReLU networks with scalar input exhibit similar behavior. In Appendix C.1 we further analyze Lipschitz classes of functions (e.g., linear functions, kernels and neural networks), and show that when the data is well-clustered, these algorithms can be teachers as well. In all cases, the sample complexity needed for the samplers or teachers is lower than the complexity required for learning.

**Nearest Neighbour Algorithm**

For some set $S = \{(\mathbf{x}_1, y_1), \ldots, (\mathbf{x}_m, y_m)\} \subseteq \mathcal{X} \times \mathcal{Y}$ we denote $S_{\mathcal{X}} = \{\mathbf{x}_1, \ldots, \mathbf{x}_m\}$. Fix some metric $d$ over the space $\mathcal{X}$. In this section, we assume that the metric space $(\mathcal{X}, d)$ satisfies the Heine-Borel property—that is, every closed and bounded set in $\mathcal{X}$ is compact[4]. For some finite set $S \subseteq \mathcal{X} \times \mathcal{Y}$, and some $\mathbf{x} \in \mathcal{X}$, denote $d(\mathbf{x}, S) := \min_{\mathbf{x}' \in S_{\mathcal{X}}} d(\mathbf{x}, \mathbf{x}')$ and $\pi(\mathbf{x}, S) := \arg\min_{(\mathbf{x}', y') \in S} d(\mathbf{x}, \mathbf{x}')$. Additionally, we define the set k-$\pi(\mathbf{x}, S_{\mathcal{X}}) \subseteq S$ to be the set of $k$ points in $S$ that are closest to $\mathbf{x}$. That is, k-$\pi(\mathbf{x}, S)$ is a set of size $k$ s.t. for any $(\mathbf{x}', y') \in$ k-$\pi(\mathbf{x}, S)$ and $(\tilde{\mathbf{x}}, \tilde{y}) \in S \setminus$ k-$\pi(\mathbf{x}, S)$ it holds that $d(\mathbf{x}, \mathbf{x}') \leq d(\mathbf{x}, \tilde{\mathbf{x}})$[5]. For some distribution $\mathcal{D}$, we say that $\mathcal{D}$ is $\lambda$-Lipschitz if for all $\mathbf{x}, \mathbf{x}' \in \mathrm{supp}(\mathcal{D}_{\mathcal{X}})$ and $y \in \mathcal{Y}$ it holds that $|\mathbb{P}_{\mathcal{D}}[y|\mathbf{x}] - \mathbb{P}_{\mathcal{D}}[y|\mathbf{x}']| \leq \lambda d(\mathbf{x}, \mathbf{x}')$.

For some odd $k \geq 1$, define $\mathcal{A}_{\text{k-NN}}(S)(\mathbf{x}) := \arg\max_{\hat{y} \in \mathcal{Y}} |(\mathbf{x}', y') \in$ k-$\pi(\mathbf{x}, S), y' = \hat{y}|$, i.e., $\mathcal{A}_{\text{k-NN}}(S)$ is the k-NN algorithm over sample $S$. We start by showing that the $\mathcal{A}_{\text{1-NN}}$ is a sampler. The distributional sample complexity of $\mathcal{A}_{\text{1-NN}}$ depends on the number of $\epsilon$-balls that can cover a $1 - \delta$ mass of the distribution (Lemma 23 in the Appendix shows that this number is always finite). Given such cover, we can guarantee that with a large enough sample, we find a candidate example in each of the balls that have non-negligible mass. In that case, if the distribution of labels does not change significantly in each ball, $\mathcal{A}_{\text{1-NN}}$ is indeed a sampler:

**Theorem 13.** *Let $\mathcal{D}$ be some $\lambda$-Lipschitz distribution. Then, $\mathcal{A}_{\text{1-NN}}$ is a **sampler** for $\mathcal{D}$.*

Next, we study the k-NN algorithm for any odd $k \geq 1$. Similarly to the analysis of the 1-NN case, we show that a large enough sample guarantees at least $k$ candidates in each of the $\epsilon$-ball covering the distribution. In this case, the prediction of the k-NN algorithm is the majority vote over the $k$ neighbours in each ball. This will get closer to the Bayes optimal prediction as $k$ grows, and therefore the k-NN algorithm is not a sampler for $k > 1$. However, we show that it is always a teacher:

**Theorem 14.** *Let $\mathcal{D}$ be some $\lambda$-Lipschitz distribution. Then, $\mathcal{A}_{\text{k-NN}}$ is a **teacher** for $\mathcal{D}$.*

The core argument for proving Theorem 14 relies on using a variant of Condorcet's Jury Theorem (CJT). Roughly speaking, CJT states that the accuracy of the majority vote of a set of predictors is better than the average accuracy of the individual predictors. In the k-NN case, each of the $k$ candidates casts a "vote", and using CJT we show that this improves over the 1-NN prediction, which is already a sampler (and hence a teacher). Notice that this works for any $k \geq 1$, and we do not need to require $k$ to be large enough, as would be required in order to get a learner.

**Bounded-Norm Infinite-Width ReLU Networks**

We continue with investigating infinite-widths neural-network with weights of bounded norm, following the setting studied in [24, 22]. We define a ReLU network of width $k$ and depth 2 by $h_\theta(\mathbf{x}) = \sum_{i=1}^{k} w_i^{(2)} \sigma\left(\left\langle \mathbf{w}_i^{(1)}, \mathbf{x}\right\rangle + b_i^{(1)}\right) + b^{(2)}$, where $\theta = (k, W^{(1)}, W^{(2)}, b^{(1)}, b^{(2)})$, and $\sigma$ is the ReLU activation. As in [24], we consider the Euclidean norm of non-biased weights: $C(\theta) = \frac{1}{2} \sum_{i=1}^{k} \left(\left(w_i^{(2)}\right)^2 + \left\|\mathbf{w}_i^{(1)}\right\|_2^2\right)$. Now, consider fitting a sample $S \subseteq \mathcal{X} \times \mathcal{Y}$ with a network

---

[4]Specifically, the Heine-Borel property holds for $\mathbb{R}^n$ where $d$ is induced by some norm.

[5]If there are multiple choices for such set, we choose one arbitrarily.

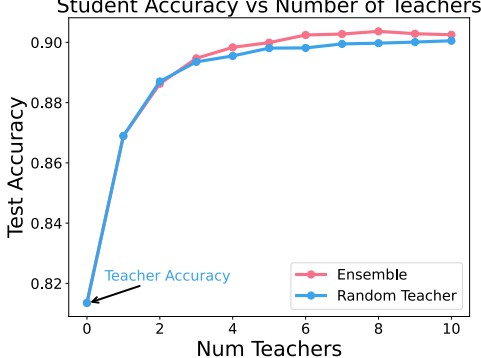

| Experiment | Test Accuracy $\pm$ std |
|---|---|
| One Teacher | 0.868 $\pm$5e-3 |
| 5 Random Teachers | 0.898 $\pm$2e-3 |
| 10 Random Teachers | 0.900 $\pm$2e-3 |
| 5 Teacher Ensemble | 0.899 $\pm$2e-3 |
| 10 Teacher Ensemble | 0.902 $\pm$1e-3 |
| 10-Ensemble Inference | 0.878 |
| 10-Teacher Clean Ens. | 0.934 $\pm$0.8e-3 |
| Teacher Accuracy | 0.813 $\pm$4.7e-2 |

Figure 2: The effect of the number of teachers on the performance of the student. We also include the teacher's accuracy marked as 0 teachers.

Table 1: Comparison of teachers, students and ensembles test performance. Note that both ensembling and distillation reduce the effect of noise.

$h_\theta$ with $C(\theta)$ acting as regularization. Namely, observe the following objective function:

$$R(S) = \inf_\theta C(\theta) \text{ s.t. } h_\theta(\mathbf{x}) = y \text{ for all } (\mathbf{x}, y) \in S$$

In the one-dimensional case, i.e. when $\mathcal{X} = \mathbb{R}$, [24] shows that $R(S)$ gives the linear spline interpolation of the data points. Using this, we show that ReLU networks in this setting are samplers:

**Theorem 15.** *For algorithm $\mathcal{A}$ that takes a sample $S$ and returns $h$ s.t. $h(x) = \text{sign}\left(h_{\hat{\theta}}(x)\right)$, for $\hat{\theta} = R(S)$. Let $\mathcal{D}$ be some continuous $\lambda$-Lipschitz distribution. Then, $\mathcal{A}$ is a **sampler** for $\mathcal{D}$.*

The above shows that when we don't limit the size of the network and use the regularization $C(\theta)$, the resulting algorithm is a sampler. Observe that if we do not introduce regularization, one can construct a ReLU network $h_\theta$ that outputs a constant value (e.g., 1) for all $x \in \mathbb{R}$, outside of infinitesimally small neighbourhoods of the points of $S$, where $h_\theta$ interpolates the data (namely, $h_\theta$ is constant with very narrow "spikes" towards the correct labels of the examples in the sample). Thus, on new points $h_\theta$ evaluates to 1 with high probability, so it does not behave like a sampler.

Admittedly, going beyond the one-dimensional case is more challenging, as it requires understanding the high-dimensional geometry of the function returned by $R(S)$. While we defer this case for future work, we note that the analysis of [22] gives some technical tools for understanding the multivariate version of the above problem. Specifically, [22] show that solving $R(S)$ is equivalent to minimizing a specific norm in function-space, which controls the complexity of the learned function. Among other things, the authors show that controlling this norm prevents a "spiking" behavior as described above.

## 5    Experiments

To this point, we saw that getting a sampler (and thus a teacher) from a noisy distribution can be more sample efficient than getting a learner. Furthermore, we showed that we can leverage multiple independent teachers to approximate the Bayes optimal classifier either via ensembling at inference time or via distillation on unlabeled data. We now complement our theoretical results with an experimental evaluation, showing the benefit of using distillation when training on noisy data. While in our theoretical setting we studied teachers that are trained on entirely disjoint training sets, in practice we find it more effective to train the teachers on overlapping datasets, as well as training on same dataset with different random initialization.

To get the teachers, we train a ResNet-18 [13] on CIFAR-10 with 20%-*fixed* and *non-uniform* label noise (see full details in D). We see that our teachers achieve 81.3% test accuracy (see Table 1) and behave closely to samplers (see Figure 1) reproducing the results of [20]. We now compare the three methods considered before for using teachers to get learners: 1) Test time Ensembling; 2) Ensemble as distillation teacher and; 3) Random teacher distillation. For distillation, we train a student network on the CIFAR-5m, a large (5-million examples) dataset that resembles the CIFAR-10 dataset [21], where the labels are provided by the previously trained teachers. We report our results

in Table 1, where the reported accuracies are on the CIFAR-10 test data. Observe that using an ensemble for inference reduces the noise significantly, and achieves test accuracy of $87.8\%$ (versus $81.3\%$ for a single teacher). When applying distillation, both random pseudo-labeling and ensemble pseudo-labeling further increase the test accuracy to about $90\%$. In addition, we study how the number of teachers affects performance (see Figure 2). We observe that both random pseudo-labeling and ensemble majority improve in performance when the number of teachers grow.

# 6 Acknowledgements

This work supported by Simons Investigator Fellowship, NSF grants DMS-2134157 and CCF-1565264, DARPA grant W911NF2010021,and DOE grant DE-SC0022199.

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
