# A Proofs for Section 2

*Proof.* of Lemma 3.

For every $\mathbf{x}$ denote $\gamma(\mathbf{x}) = \mathbb{P}\left[y = h^*(\mathbf{x})|\mathbf{x}\right] - \mathbb{P}\left[y \neq h^*(\mathbf{x})|\mathbf{x}\right]$, and since $h^*$ is the Bayes optimal predictor it holds that $\gamma(\mathbf{x}) \geq 0$. Observe the following:

$$
\begin{aligned}
L_{\mathcal{D}}(h) &= \mathbb{E}_{\mathbf{x}\sim\mathcal{D}}\left[\mathbb{P}\left[y \neq h(\mathbf{x})|\mathbf{x}\right]\right] \\
&= \mathbb{E}_{\mathbf{x}\sim\mathcal{D}}\left[\mathbb{P}\left[y \neq h(\mathbf{x})|\mathbf{x}\right] \cdot \mathbf{1}\{h(\mathbf{x}) = h^*(\mathbf{x})\}\right] \\
&\quad + \mathbb{E}_{\mathbf{x}\sim\mathcal{D}}\left[\mathbb{P}\left[y \neq h(\mathbf{x})|\mathbf{x}\right] \cdot \mathbf{1}\{h(\mathbf{x}) \neq h^*(\mathbf{x})\}\right] \\
&= \mathbb{E}_{\mathbf{x}\sim\mathcal{D}}\left[\mathbb{P}\left[y \neq h^*(\mathbf{x})|\mathbf{x}\right] \cdot \mathbf{1}\{h(\mathbf{x}) = h^*(\mathbf{x})\}\right] \\
&\quad + \mathbb{E}_{\mathbf{x}\sim\mathcal{D}}\left[\mathbb{P}\left[y = h^*(\mathbf{x})|\mathbf{x}\right] \cdot \mathbf{1}\{h(\mathbf{x}) \neq h^*(\mathbf{x})\}\right] \\
&= \mathbb{E}_{\mathbf{x}\sim\mathcal{D}}\left[\mathbb{P}\left[y \neq h^*(\mathbf{x})|\mathbf{x}\right] \cdot \left(\mathbf{1}\{h(\mathbf{x}) = h^*(\mathbf{x})\} + \mathbf{1}\{h(\mathbf{x}) \neq h^*(\mathbf{x})\}\right)\right] \\
&\quad + \mathbb{E}_{\mathbf{x}\sim\mathcal{D}}\left[\gamma(\mathbf{x}) \cdot \mathbf{1}\{h(\mathbf{x}) \neq h^*(\mathbf{x})\}\right] \\
&= L_{\mathcal{D}}(h^*) + \mathbb{E}_{\mathbf{x}\sim\mathcal{D}}\left[\gamma(\mathbf{x}) \cdot \mathbf{1}\{h(\mathbf{x}) \neq h^*(\mathbf{x})\}\right]
\end{aligned}
\tag{2}
$$

Now, notice that we have:

$$
\begin{aligned}
\mathbb{E}_{\mathbf{x}\sim\mathcal{D}}\left[\gamma(\mathbf{x}) \cdot \mathbf{1}\{h(\mathbf{x}) \neq h^*(\mathbf{x})\}\right] &\geq \mathbb{E}_{\mathbf{x}\sim\mathcal{D}}\left[\gamma(\mathbf{x}) \cdot \mathbf{1}\{h(\mathbf{x}) \neq h^*(\mathbf{x})\} \cdot \mathbf{1}\{\gamma(\mathbf{x}) \geq \gamma_\delta(\mathcal{D})\}\right] \\
&\geq \gamma_\delta(\mathcal{D}) \mathbb{P}_{\mathcal{D}}(A \cap B)
\end{aligned}
\tag{3}
$$

where $A$ denotes the event where $h(\mathbf{x}) \neq h^*(\mathbf{x})$ and $B$ denotes the event where $\gamma(\mathbf{x}) \geq \gamma_\delta(\mathcal{D})$. By definition of the loss we have $\mathbb{P}_{\mathcal{D}}(A) = L_{\mathcal{D}^*}(h)$, and by definition of the margin we have $\mathbb{P}_{\mathcal{D}}(B) \geq 1 - \delta$. Therefore, we have:

$$
\mathbb{P}_{\mathcal{D}}(A \cap B) = \mathbb{P}_{\mathcal{D}}(A) + \mathbb{P}_{\mathcal{D}}(B) - \mathbb{P}_{\mathcal{D}}(A \cup B) \geq L_{\mathcal{D}^*}(h) + (1 - \delta) - 1 = L_{\mathcal{D}^*}(h) - \delta
\tag{4}
$$

Now, combining Eq. (2), (3) and (4), together with the fact that $L_{\mathcal{D}}(h) \leq L_{\mathcal{D}}(h^*) + \epsilon$, we get:

$$
\gamma_\delta(\mathcal{D})(L_{\mathcal{D}^*}(h) - \delta) + L_{\mathcal{D}}(h^*) \leq L_{\mathcal{D}}(h) \leq L_{\mathcal{D}}(h^*) + \epsilon
$$

and so the required follows.

$\square$

*Proof.* of Theorem 2.

Fix some $\epsilon \in (0, 1)$ and let $\epsilon' = \frac{\epsilon\gamma(\mathcal{D})}{2}$ and $\delta' = \frac{\epsilon}{2}$. By the Fundamental Theorem of Statistical Learning (see [25]), there exists some universal constant $C$ s.t. taking $m = C\frac{\text{VC}(\mathcal{H}) + \log(1/\delta')}{(\epsilon')^2}$ we get that w.p. at least $1 - \delta'$ over sampling $S \sim \mathcal{D}^m$ it holds that:

$$
L_{\mathcal{D}}(\text{ERM}_{\mathcal{H}}(S)) \leq \inf_{h \in \mathcal{H}} L_{\mathcal{D}}(h) + \epsilon' = L_{\mathcal{D}}(f_{\mathcal{D}}^*) + \epsilon'
$$

where we use the fact that $f_{\mathcal{D}}^* \in \mathcal{H}$ is the Bayes optimal of $\mathcal{D}$. Now, from Lemma 3 it holds that, w.p. at least $1 - \delta'$ it holds that (note that $\gamma(\mathcal{D}) = \gamma_0(\mathcal{D})$),

$$
L_{\mathcal{D}^*}(\text{ERM}_{\mathcal{H}}(S)) \leq \frac{\epsilon'}{\gamma(\mathcal{D})}
$$

So, we get that:

$$
\mathbb{E}_{S\sim\mathcal{D}^m} L_{\mathcal{D}^*}(\text{ERM}_{\mathcal{H}}(S)) \leq \frac{\epsilon'}{\gamma(\mathcal{D})} + \delta' = \epsilon
$$

$\square$

*Proof.* of Lemma 5.

Let the event $E = \{(\mathbf{x}, y)|\ y \neq f^*(\mathbf{x})\}$, then,

$$
\begin{aligned}
\left|\eta(\mathcal{D}) - \mathbb{E}_{S\sim\mathcal{D}^m} L_{\mathcal{D}^*}(\mathcal{A}(S))\right| &= \left|\mathbb{P}_{\mathbf{x},y\sim\mathcal{D}}[y \neq f^*(\mathbf{x})] - \mathbb{P}_{\substack{\mathbf{x}\sim\mathcal{D}_{\mathcal{X}} \\ S\sim\mathcal{D}^m}}[\mathcal{A}(S)(\mathbf{x}) \neq f^*(\mathbf{x})]\right| = \\
&= |\mathcal{D}(E) - \mathcal{A}(\mathcal{D}^m)(E)| \leq \sup_E |\mathcal{D}(E) - \mathcal{A}(\mathcal{D}^m)(E)| = \\
&= TV(\mathcal{D}, \mathcal{A}(\mathcal{D}^m)) = \varepsilon
\end{aligned}
$$

$\square$

*Proof.* of Theorem 6.

We follow a proof similar to Chapter 28.2.1 of [25].

Let $\gamma = \sqrt{\frac{\log(4/3)}{2M}}$. For every $b \in \{\pm 1\}$, let $\mathcal{D}_b$ be the distributions concentrated on a single example $\mathbf{x} \in \mathcal{X}$, with label,

$$y \sim P_b(y) = \text{Bernoulli}\left(\frac{1+b\gamma}{2}\right) = \begin{cases} \frac{1+b\gamma}{2} & \text{if } y = 1 \\ \frac{1-b\gamma}{2} & \text{if } y = -1 \end{cases}$$

Take $\mathcal{P} = \{\mathcal{D}_+, \mathcal{D}_-\}$. Observe that the algorithm $\mathcal{A}$ that takes a single sample $(\mathbf{x}, y_0)$ and outputs $y_0$ is a sampler for every $\mathcal{D} \in \mathcal{P}$.

Let $\mathbf{y} \in \{\pm 1\}^m$ be the sequence of labels observed by the algorithm $\mathcal{A}$, and denote by $\mathcal{A}(\mathbf{y}) \in \{\pm 1\}$ the label that $\mathcal{A}$ outputs for $\mathbf{x}$ when observing the sequence of labels $\mathbf{y}$. Note, that $\mathcal{D}^*$ will be a *constant distribution* concentrated on $(\mathbf{x}, b)$. Therefore, we have:

$$\mathop{\mathbb{E}}_{S \sim \mathcal{D}_b^m} L_{\mathcal{D}^*}(\mathcal{A}(S)) = \mathop{\mathbb{E}}_{S \sim \mathcal{D}_b^m} \mathbf{1}\{\mathcal{A}(S)(\mathbf{x}) \neq b\} = \mathop{\mathbb{E}}_{\mathbf{y} \sim P_b^m} \mathbf{1}\{\mathcal{A}(\mathbf{y}) \neq b\}$$

Denote $N_+ := \{\mathbf{y} \in \{\pm 1\}^m : \sum_i y_i \geq 0\}$ and $N_- = \{\pm 1\}^m \setminus N_+$. Then:

$$\mathop{\mathbb{E}}_{\mathbf{y} \sim P_+^m} \mathbf{1}\{\mathcal{A}(\mathbf{y}) = -1\} + \mathop{\mathbb{E}}_{\mathbf{y} \sim P_-^m} \mathbf{1}\{\mathcal{A}(\mathbf{y}) = 1\}$$

$$= \sum_{\mathbf{y}} P_+(\mathbf{y})\mathbf{1}\{\mathcal{A}(\mathbf{y}) = -1\} + P_-(\mathbf{y})\mathbf{1}\{\mathcal{A}(\mathbf{y}) = 1\}$$

$$= \sum_{\mathbf{y} \in N_+} P_+(\mathbf{y})\mathbf{1}\{\mathcal{A}(\mathbf{y}) = -1\} + P_-(\mathbf{y})\mathbf{1}\{\mathcal{A}(\mathbf{y}) = 1\}$$

$$+ \sum_{\mathbf{y} \in N_-} P_+(\mathbf{y})\mathbf{1}\{\mathcal{A}(\mathbf{y}) = -1\} + P_-(\mathbf{y})\mathbf{1}\{\mathcal{A}(\mathbf{y}) = 1\}$$

$$\geq \sum_{\mathbf{y} \in N_+} P_-(\mathbf{y}) + \sum_{\mathbf{y} \in N_-} P_+(\mathbf{y}) \geq \frac{1}{2}\left(1 - \sqrt{1 - \exp(-2m\gamma^2)}\right)$$

where the last inequality follows from Lemma B.11 in [25]. So, if $m < \frac{\log(4/3)}{2\gamma^2} = M$ we get:

$$\mathop{\mathbb{E}}_b \mathop{\mathbb{E}}_{S \sim \mathcal{D}_b^m} L_{\mathcal{D}^*}(\mathcal{A}(S)) = \frac{1}{2}\left(\mathop{\mathbb{E}}_{\mathbf{y} \sim P_+^m} \mathbf{1}\{\mathcal{A}(\mathbf{y}) = -1\} + \mathop{\mathbb{E}}_{\mathbf{y} \sim P_-^m} \mathbf{1}\{\mathcal{A}(\mathbf{y}) = 1\}\right) > \frac{1}{8}$$

and we get there exists $\mathcal{D} \in \mathcal{P}$ s.t. if $m < M$ then $\mathbb{E}_{S \sim \mathcal{D}^m} L_{\mathcal{D}^*}(\mathcal{A}(S)) > \frac{1}{8}$. $\qquad\square$

*Proof.* of Theorem 8.

To see property *1.* of Definition 7, we show that if two distributions over $(\mathbf{x}, y)$ are close in total variation, then the Bayes optimal classifier for both has to be similar. That is,

$$\text{TV}(\mathcal{D}, \mathcal{D}') < \varepsilon \implies \mathbb{P}_{\mathbf{x} \sim \mathcal{D}}[f_{\mathcal{D}}^*(\mathbf{x}) \neq f_{\mathcal{D}'}^*(\mathbf{x})] \leq \varepsilon/\gamma$$

Note, for $\mathbf{x} \sim \mathcal{D}_{\mathcal{X}}$ we have $y_{\mathbf{x}} =: f_{\mathcal{D}}^*(\mathbf{x}) \neq f_{\mathcal{D}'}^*(\mathbf{x}) =: \hat{y}_{\mathbf{x}}$ if and only if $\mathbb{P}_{\mathcal{D}'}[y_{\mathbf{x}}|\mathbf{x}] < \mathbb{P}_{\mathcal{D}'}[\hat{y}_{\mathbf{x}}|\mathbf{x}]$, but the margin condition guarantees that $\mathbb{P}_{\mathcal{D}}[y_{\mathbf{x}}|\mathbf{x}] - \mathbb{P}_{\mathcal{D}}[\hat{y}_{\mathbf{x}}|\mathbf{x}] \geq \gamma$, thus,

$$\mathop{\mathbb{P}}_{\mathbf{x} \sim \mathcal{D}}[f_{\mathcal{D}}^*(\mathbf{x}) \neq f_{\mathcal{D}'}^*(\mathbf{x})] \leq \mathop{\mathbb{P}}_{\mathbf{x} \sim \mathcal{D}_{\mathcal{X}}}[|\mathbb{P}_{\mathcal{D}}[y_{\mathbf{x}}|\mathbf{x}] - \mathbb{P}_{\mathcal{D}'}[y_{\mathbf{x}}|\mathbf{x}]| > \gamma] \leq$$

$$\leq \mathop{\mathbb{E}}_{\mathbf{x} \sim \mathcal{D}}[|\mathbb{P}_{\mathcal{D}}[y_{\mathbf{x}}|\mathbf{x}] - \mathbb{P}_{\mathcal{D}'}[y_{\mathbf{x}}|\mathbf{x}]|]/\gamma.$$

Where we use Markov inequality for the second transition. Now, we can use the alternative definition of TV to conclude the proof (here $\mathbf{p}(\mathbf{x})$ is the Radon–Nikodym measure of $\mathbf{x}$ under the marginal $\mathcal{D}_{\mathcal{X}}$ and $\mathbf{p}_{\mathcal{D}}(\mathbf{x}, y)$ is the Radon–Nikodym measure of $(\mathbf{x}, y)$ under $\mathcal{D}$):

$$\mathop{\mathbb{E}}_{\mathbf{x} \sim \mathcal{D}_{\mathcal{X}}}[|\mathbb{P}_{\mathcal{D}}[y_{\mathbf{x}}|\mathbf{x}] - \mathbb{P}_{\mathcal{D}'}[y_{\mathbf{x}}|\mathbf{x}]|] = \int |\mathbb{P}_{\mathcal{D}}[y_{\mathbf{x}}|\mathbf{x}] - \mathbb{P}_{\mathcal{D}'}[y_{\mathbf{x}}|\mathbf{x}]|\, \mathbf{p}(\mathbf{x}) \leq$$

$$\leq \frac{1}{2} \int |\mathbf{p}_{\mathcal{D}}(\mathbf{x}, y) - \mathbf{p}_{\mathcal{D}'}(\mathbf{x}, y)| \leq \varepsilon$$

Where the penultimate inequality is based on an easy corollary of the triangle inequality: $\forall y$ we have

$$\sum_y |\mathbb{P}_{\mathcal{D}}[y|\mathbf{x}] - \mathbb{P}_{\mathcal{D}'}[y|\mathbf{x}]| \geq 2|\mathbb{P}_{\mathcal{D}}[y|\mathbf{x}] - \mathbb{P}_{\mathcal{D}'}[y|\mathbf{x}]|.$$

We proceed to prove the 2$^{\text{rd}}$ property. For each $\mathbf{x} \in \mathcal{X}$ let $y_1$ and $y_2$ be the two most likely labels respectively with respect to the distribution $\mathcal{D}$, that is, $y_1(\mathbf{x}) = \arg\max_y \mathbb{P}_{\mathcal{D}}[y|\mathbf{x}]$ and $y_2(\mathbf{x}) = \arg\max_{y \neq y_1(\mathbf{x})} \mathbb{P}_{\mathcal{D}}[y|\mathbf{x}]$ and $y_1', y_2'$ defined similarly for $\mathcal{D}'$. Then, for a given $\mathbf{x}$, if the margin is small, i.e., $\mathbb{P}_{\mathcal{D}'}[y_1'|\mathbf{x}] - \mathbb{P}_{\mathcal{D}'}[y_2'|\mathbf{x}] < \gamma - \tau$ then we will want to prove that the following holds:

$$\sum_y |\mathbb{P}_{\mathcal{D}}[y|\mathbf{x}] - \mathbb{P}_{\mathcal{D}'}[y|\mathbf{x}]| > \tau \tag{5}$$

If $y_1 \neq y_1'$ then with probability 1 we have $\mathbb{P}_{\mathcal{D}}[y_1|\mathbf{x}] - \mathbb{P}_{\mathcal{D}}[y_1'|\mathbf{x}] > \gamma$. Using the definition of $y_1'$,

$$\mathbb{P}_{\mathcal{D}}[y_1|\mathbf{x}] - \mathbb{P}_{\mathcal{D}}[y_1'|\mathbf{x}] + \mathbb{P}_{\mathcal{D}'}[y_1'|\mathbf{x}] - \mathbb{P}_{\mathcal{D}'}[y_1|\mathbf{x}] > \gamma > \tau$$

If, on the other hand, $y_1 = y_1'$ using $\mathbb{P}_{\mathcal{D}}[y_1|\mathbf{x}] - \mathbb{P}_{\mathcal{D}}[y_2|\mathbf{x}] > \gamma$ again we have (by summing up the inequalities):

$$\mathbb{P}_{\mathcal{D}'}[y_1'|\mathbf{x}] - \mathbb{P}_{\mathcal{D}'}[y_2'|\mathbf{x}] + \mathbb{P}_{\mathcal{D}'}[y_2'|\mathbf{x}] - \mathbb{P}_{\mathcal{D}'}[y_1'|\mathbf{x}] > \tau$$

So in both cases Equation 5 holds. Thus,

$$\mathbb{P}_{\mathbf{x}}[\mathbb{P}_{\mathcal{D}'}[y_1'(\mathbf{x})|\mathbf{x}] - \mathbb{P}_{\mathcal{D}'}[y_2'(\mathbf{x})|\mathbf{x}] < \gamma - \tau] \leq$$

$$\mathbb{P}_{\mathbf{x}}\left[\sum_y |\mathbb{P}_{\mathcal{D}}[y|\mathbf{x}] - \mathbb{P}_{\mathcal{D}'}[y|\mathbf{x}]| > \tau\right] \leq$$

$$\mathbb{E}_{\mathbf{x}}\left[\sum_y |\mathbb{P}_{\mathcal{D}}[y|\mathbf{x}] - \mathbb{P}_{\mathcal{D}'}[y|\mathbf{x}]|\right] / \tau = 2\text{TV}(\mathcal{D}, \mathcal{D}')/\tau = 2\frac{\varepsilon}{\tau}$$

$\square$

*Proof.* of Theorem 9.

Fix $\epsilon \in (0, 1)$ and $0 < \tau < \gamma(\mathcal{D})$. Let $m = m\left(\frac{\epsilon(1-\gamma(\mathcal{D})+\tau)}{2}\right)$. Fix some $\mathbf{x} \in \mathcal{X}$ such that

$$f_{\mathcal{D}}^*(\mathbf{x}) \neq f_{\mathcal{A}(\mathcal{D}^m)}^*(\mathbf{x}) = \arg\max_y \mathbb{P}_{\mathcal{A}(\mathcal{D}^m)}[y|\mathbf{x}]$$

Then,

$$\mathbb{P}_{S \sim \mathcal{D}^m}[\mathcal{A}(S)(\mathbf{x}) \neq f_{\mathcal{D}}^*(\mathbf{x})] = \mathbb{P}_{(\mathbf{x},y) \sim \mathcal{A}(\mathcal{D}^m)}[y \neq f_{\mathcal{D}}^*(\mathbf{x})|\mathbf{x}] \geq \frac{1}{2}$$

Therefore, since $\mathcal{A}$ is a learner with sample complexity $m(\cdot)$ we have:

$$\frac{\epsilon}{2} \geq \mathbb{E}_{S \sim \mathcal{D}^m} L_{\mathcal{D}^*}(\mathcal{A}(S)) = \mathbb{E}_{\mathbf{x} \sim \mathcal{D}_{\mathcal{X}}} \mathbb{P}_{S \sim \mathcal{D}^m}[\mathcal{A}(S)(\mathbf{x}) \neq f_{\mathcal{D}}^*(\mathbf{x})]$$

$$\geq \mathbb{E}_{\mathbf{x}}\left[\mathbb{P}_S[\mathcal{A}(S)(\mathbf{x}) \neq f_{\mathcal{D}}^*(\mathbf{x})]\Big| f_{\mathcal{D}}^*(\mathbf{x}) \neq f_{\mathcal{A}(\mathcal{D}^m)}^*(\mathbf{x})\right] \cdot \mathbb{P}_{\mathbf{x}}\left[f_{\mathcal{D}}^*(\mathbf{x}) \neq f_{\mathcal{A}(\mathcal{D}^m)}^*(\mathbf{x})\right]$$

$$\geq \frac{1}{2}\mathbb{P}_{\mathbf{x}}\left[f_{\mathcal{D}}^*(\mathbf{x}) \neq f_{\mathcal{A}(\mathcal{D}^m)}^*(\mathbf{x})\right] = \frac{1}{2}L_{\mathcal{D}^*}\left(f_{\mathcal{A}(\mathcal{D}^m)}^*\right)$$

So, the first condition of Definition 7 holds. For the second condition, observe that since $f_{\mathcal{A}(\mathcal{D}^m)}^*$ is the Bayes optimal classifier, we have:

$$\eta(\mathcal{A}(\mathcal{D}^m)) = \mathbb{P}_{(\mathbf{x},y) \sim \mathcal{A}(\mathcal{D}^m)}\left[y \neq f_{\mathcal{A}(\mathcal{D}^m)}^*(\mathbf{x})\right] \leq \mathbb{P}_{\mathcal{A}(\mathcal{D}^m)}[y \neq f_{\mathcal{D}}^*(\mathbf{x})]$$

$$= \mathbb{E}_{\mathbf{x} \sim \mathcal{D}} \mathbb{P}_{S \sim \mathcal{D}^m}[\mathcal{A}(S)(\mathbf{x}) \neq f_{\mathcal{D}}^*(\mathbf{x})] = \mathbb{E}_{S \sim \mathcal{D}^m} L_{\mathcal{D}^*}(\mathcal{A}(S)) \leq \frac{\delta(1-\gamma(\mathcal{D})+\tau)}{2}$$

where the last inequality is using the fact that $\mathcal{A}$ is a learner. From Lemma 16, since $\eta(\mathcal{A}(\mathcal{D})) \leq \frac{\delta(1-\gamma(\mathcal{D})+\tau)}{2}$, it holds that $\gamma_\delta(\mathcal{A}(\mathcal{D})) \geq \gamma(\mathcal{D}) - \tau$. $\square$

**Lemma 16.** *Let $\mathcal{D}$ be a distribution with $\mu$-bounded noise, i.e., $\eta(\mathcal{D}) = \mathbb{P}_{\mathbf{x},y\sim\mathcal{D}}[y \neq f^*(\mathbf{x})] \leq \mu$ where $f^*$ is the Bayes optimal classifier. Let $0 < \gamma < 1$ be some positive constant denoting a margin. Then,*

$$\mathop{\mathbb{P}}_{\mathbf{x}\sim\mathcal{D}_{\mathcal{X}}} \left[ \mathbb{P}_{\mathcal{D}}(f_{\mathcal{D}}^*(\mathbf{x})|\mathbf{x}) < \max_{y\neq f_{\mathcal{D}}^*(\mathbf{x})} \mathbb{P}_{\mathcal{D}}(y|\mathbf{x}) + \gamma \right] \leq \frac{2\cdot\mu}{(1-\gamma)}$$

*Proof.* For each $\mathbf{x} \in \mathcal{X}$ let $y_1(\mathbf{x})$ and $y_2(\mathbf{x})$ be the two most likely labels respectively, that is, $y_1(\mathbf{x}) = \arg\max_y \mathbb{P}[y|\mathbf{x}]$ and $y_2(\mathbf{x}) = \arg\max_{y\neq y_1(\mathbf{x})} \mathbb{P}[y|\mathbf{x}]$. Let $\gamma_{\mathbf{x}} = \mathbb{P}[y_1(\mathbf{x})|\mathbf{x}] - \mathbb{P}[y_2(\mathbf{x})|\mathbf{x}]$ and denote the set of small margin examples $B = \{\mathbf{x}|\gamma_{\mathbf{x}} \leq \gamma\}$. Then we have,

$$\eta(\mathcal{D}) = \mathbb{E}_{\mathbf{x}}[\mathbb{P}[Y \neq y_1(\mathbf{x})|\mathbf{x}]] \geq$$
$$\geq \mathbb{E}_{\mathbf{x}}[\mathbb{P}[Y \neq y_1(\mathbf{x})|\mathbf{x}]|\mathbf{x} \in B]\,\mathbb{P}[\mathbf{x} \in B] \geq$$
$$\geq \mathbb{P}(B) \cdot \frac{1-\gamma}{2}$$

Where the last inequality is proven via the following lemma: $\qquad\square$

**Lemma 17.** *Given a fixed $\mathbf{x} \in B$ s.t. $\gamma_{\mathbf{x}} \leq \gamma$ (in notation of Lemma 16) for some $0 < \gamma < 1$. Then,*

$$\mathbb{P}[Y \neq y_1(\mathbf{x})|\mathbf{x}] \geq \frac{1-\gamma}{2}$$

*Proof.* Since the $\mathbf{x}$ is fixed we drop all $\mathbf{x}$ related notation WLOG:

$$\mathbb{P}[Y = y_1] = 1 - \mathbb{P}[Y \neq y_1] \leq$$
$$\leq 1 - \mathbb{P}[Y = y_2] \leq$$
$$\leq 1 - \mathbb{P}[Y = y_1] + \gamma$$

Thus, by rearranging we get $\mathbb{P}[Y = y_1] \leq \frac{1+\gamma}{2}$ which implies $\mathbb{P}[Y \neq y_1] \geq \frac{1-\gamma}{2}$ $\qquad\square$

# B   Proofs of Section 3

To prove Theorem 10 we use the following Lemma:

**Lemma 18.** *Let $\mathcal{A}$ be some learning algorithm. Fix some $\gamma > 0$ and $\tau < \gamma$. Then, for every $\mathbf{x}$ s.t.*

- $\mathbb{P}_{\mathcal{A}(\mathcal{D}^m)}\left[f_{\mathcal{A}(\mathcal{D}^m)}^*(\mathbf{x}) \mid \mathbf{x}\right] > \mathbb{P}_{\mathcal{A}(\mathcal{D}^m)}\left[-f_{\mathcal{A}(\mathcal{D}^m)}^*(\mathbf{x}) \mid \mathbf{x}\right] + \gamma$ *and*

- $f_{\mathcal{A}(\mathcal{D}^m)}^*(\mathbf{x}) = f_{\mathcal{D}}^*(\mathbf{x})$

*it holds that:*

$$\mathbb{P}_{S_1,\dots,S_k\sim\mathcal{D}^m}\left[f_{\mathcal{D}}^*(\mathbf{x})\frac{1}{k}\sum_{i=1}^{k}\mathcal{A}(S_i)(\mathbf{x}) \leq \tau\right] \leq \exp\left(-\frac{k(\gamma-\tau)^2}{4}\right)$$

*Proof.* of Lemma 18.

Fix some $\mathbf{x} \in \mathcal{X}$ and denote

$$p_{\mathbf{x}}(y) = \mathbb{P}_{\mathcal{A}(\mathcal{D}^m)}[y|\mathbf{x}] = \mathbb{P}_{S\sim\mathcal{D}^m}[\mathcal{A}(S)(\mathbf{x}) = y]$$

Let $y_{\mathbf{x}}^* = \arg\max_y p_{\mathbf{x}}(y) = f_{\mathcal{A}(\mathcal{D}^m)}^*(\mathbf{x})$. So, assume that $\mathbf{x}$ satisfies the assumption, namely assume that $p_{\mathbf{x}}(y_{\mathbf{x}}^*) > p_{\mathbf{x}}(-y_{\mathbf{x}}^*) + \gamma$ and $y_{\mathbf{x}}^* = f_{\mathcal{D}}^*(\mathbf{x})$.

Denote $y_{\mathbf{x}}^{(i)} = \mathcal{A}(S_i)(\mathbf{x})$, the prediction of the $i$-th teacher on $\mathbf{x}$. Then,

$$\mathbb{E}\left[\frac{1}{k}y_{\mathbf{x}}^*\sum_{i=1}^{k}y_{\mathbf{x}}^{(i)}\right] = \mathbb{E}\left[y_{\mathbf{x}}^*y_{\mathbf{x}}^{(1)}\right] = p_{\mathbf{x}}(y_{\mathbf{x}}^*) - p_{\mathbf{x}}(-y_{\mathbf{x}}^*) > \gamma$$

By Hoeffding's inequality we get:

$$\mathbb{P}\left[\frac{1}{k}y_{\mathbf{x}}^*\sum_{i=1}^{k}y_{\mathbf{x}}^{(i)} \leq \tau\right] \leq \exp\left(-\frac{k\left(\mathbb{E}\left[\frac{1}{k}y_{\mathbf{x}}^*\sum_{i=1}^{k}y_{\mathbf{x}}^{(i)}\right]-\tau\right)^2}{4}\right) = \exp\left(-\frac{k(\gamma-\tau)^2}{4}\right)$$

$\square$

*Proof.* of Theorem 10

Let $\mathcal{X}' \subseteq \mathcal{X}$ be the subset of points $\mathbf{x} \in \mathcal{X}$ satisfying the assumptions of Lemma 18 with $\gamma = \frac{\gamma(\mathcal{D})}{2}$ and $\tau = 0$. Observe that, using the union bound, and the properties of the teacher $\mathcal{A}$:

$$\mathbb{P}_{\mathbf{x}\sim\mathcal{D}}\left[\mathbf{x} \notin \mathcal{X}'\right]$$
$$\leq \mathbb{P}_{\mathbf{x}\sim\mathcal{D}}\left[\mathbb{P}_{\mathcal{A}(\mathcal{D}^m)}\left[f_{\mathcal{A}(\mathcal{D}^m)}^*(\mathbf{x}) \mid \mathbf{x}\right] > \mathbb{P}_{\mathcal{A}(\mathcal{D}^m)}\left[-f_{\mathcal{A}(\mathcal{D}^m)}^*(\mathbf{x}) \mid \mathbf{x}\right] + \gamma\right]$$
$$+ \mathbb{P}_{\mathbf{x}\sim\mathcal{D}}\left[f_{\mathcal{A}(\mathcal{D}^m)}^*(\mathbf{x}) \neq f_{\mathcal{D}}^*(\mathbf{x})\right]$$
$$\leq \epsilon/3 + L_{\mathcal{D}^*}\left(f_{\mathcal{A}(\mathcal{D}^m)}^*\right) \leq \frac{2\epsilon}{3}$$

Now, fix some $\mathbf{x} \in \mathcal{X}'$, and from Lemma 18 we have:

$$\mathbb{E}_{S_1,\ldots,S_k\sim\mathcal{D}^m} \mathbf{1}\{\mathcal{A}_{\mathrm{ens}}(S_1,\ldots,S_k)(\mathbf{x}) \neq f_{\mathcal{D}}^*(\mathbf{x})\} \leq \exp\left(-\frac{k\gamma^2}{4}\right) \leq \epsilon/3$$

Finally, we get:

$$\mathbb{E}_{S_1,\ldots,S_k\sim\mathcal{D}^m} L_{\mathcal{D}^*}(\mathcal{A}_{\mathrm{ens}}(S_1,\ldots,S_k))$$
$$= \mathbb{E}_{S_1,\ldots,S_k\sim\mathcal{D}^m} \mathbb{E}_{\mathbf{x}} \mathbf{1}\{\mathcal{A}_{\mathrm{ens}}(S_1,\ldots,S_k)(\mathbf{x}) \neq f_{\mathcal{D}}^*(\mathbf{x})\}$$
$$= \mathbb{P}_{\mathbf{x}\sim\mathcal{D}}\left[\mathbf{x} \in \mathcal{X}'\right] \cdot \mathbb{E}_{\mathbf{x}|\mathbf{x}\in\mathcal{X}'} \mathbb{E}_{S_1,\ldots,S_k\sim\mathcal{D}^m} \mathbf{1}\{\mathcal{A}_{\mathrm{ens}}(S_1,\ldots,S_k)(\mathbf{x}) \neq f_{\mathcal{D}}^*(\mathbf{x})\}$$
$$+ \mathbb{P}_{\mathbf{x}\sim\mathcal{D}}\left[\mathbf{x} \notin \mathcal{X}'\right] \cdot \mathbb{E}_{\mathbf{x}|\mathbf{x}\notin\mathcal{X}'} \mathbb{E}_{S_1,\ldots,S_k\sim\mathcal{D}^m} \mathbf{1}\{\mathcal{A}_{\mathrm{ens}}(S_1,\ldots,S_k)(\mathbf{x}) \neq f_{\mathcal{D}}^*(\mathbf{x})\}$$
$$\leq \left(1 - \frac{2\epsilon}{3}\right)\frac{\epsilon}{3} + \frac{2\epsilon}{3} \leq \epsilon$$

$\square$

*Proof.* of Theorem 11

Fix a sequence of $k$ subsets of examples $\mathcal{S} = (S_1,\ldots,S_k)$, and let $\widetilde{\mathcal{D}}_\mathcal{S}$ be the distribution given by sampling $\mathbf{x} \sim \mathcal{D}$ and returning $(\mathbf{x},y)$ where $y = \mathcal{A}_{\mathrm{ens}}(S_1,\ldots,S_k)(\mathbf{x})$. Let $\tilde{S}_\mathcal{S}$ be an i.i.d. sample of size $m'$ from $\widetilde{\mathcal{D}}_\mathcal{S}$. Let $h_\mathcal{S} = \mathrm{ERM}_\mathcal{H}(\widetilde{S}_\mathcal{S})$. By the Fundamental Theorem of Statistical Learning (e.g. Theorem 6.8 in [25]) w.p. at least $1 - \epsilon/4$ over sampling $\widetilde{S}_\mathcal{S}$ we have:

$$L_{\widetilde{\mathcal{D}}_\mathcal{S}}(h_\mathcal{S}) \leq \inf_{h\in\mathcal{H}} L_{\widetilde{\mathcal{D}}_\mathcal{S}}(h) + \epsilon/4 \leq L_{\widetilde{\mathcal{D}}_\mathcal{S}}(f_{\mathcal{D}}^*) + \epsilon/4$$
$$= \mathbb{P}_{\mathbf{x}\sim\mathcal{D}}\left[\mathcal{A}_{\mathrm{ens}}(\mathcal{S})(\mathbf{x}) \neq f_{\mathcal{D}}^*(\mathbf{x})\right] + \epsilon/4 = L_{\mathcal{D}^*}(\mathcal{A}_{\mathrm{ens}}(\mathcal{S})) + \epsilon/4$$

On the other hand, observe that for all $h$:

$$L_{\mathcal{D}^*}(h) = \mathbb{E}_{\mathbf{x}\sim\mathcal{D}} \mathbf{1}\{h(\mathbf{x}) \neq f_{\mathcal{D}}^*(\mathbf{x})\}$$
$$\leq \mathbb{E}_{\mathbf{x}\sim\mathcal{D}}\left(\mathbf{1}\{h(\mathbf{x}) \neq \mathcal{A}_{\mathrm{ens}}(\mathcal{S})(\mathbf{x})\} + \mathbf{1}\{\mathcal{A}_{\mathrm{ens}}(\mathcal{S})(\mathbf{x}) \neq f_{\mathcal{D}}^*(\mathbf{x})\}\right)$$
$$= L_{\widetilde{\mathcal{D}}_\mathcal{S}}(h) + L_{\mathcal{D}^*}(\mathcal{A}_{\mathrm{ens}}(\mathcal{S}))$$

Overall we get that w.p. at least $1 - \epsilon/4$ over sampling $\widetilde{S}_\mathcal{S}$ we have:

$$L_{\mathcal{D}^*}(h_\mathcal{S}) \leq 2L_{\mathcal{D}^*}(\mathcal{A}_{\mathrm{ens}}(\mathcal{S})) + \epsilon/4$$

and therefore:
$$\mathop{\mathbb{E}}_{\widetilde{\mathcal{S}}_{\mathcal{S}} \sim \widetilde{\mathcal{D}}_{\mathcal{S}}^{m'}} L_{\mathcal{D}^*}(h_{\mathcal{S}}) \leq 2L_{\mathcal{D}^*}(\mathcal{A}_{\mathrm{ens}}(\mathcal{S})) + \epsilon/2$$

Finally, using Theorem 10 we get:
$$\mathop{\mathbb{E}}_{S_1,\ldots,S_k,\widetilde{S}} L_{\mathcal{D}^*}(h) \leq 2 \mathop{\mathbb{E}}_{S_1,\ldots,S_k \sim \mathcal{D}^m} L_{\mathcal{D}^*}(\mathcal{A}_{\mathrm{ens}}(S_1,\ldots,S_k)) + \epsilon/2 \leq \epsilon$$

where $h$ is the output of the Ensemble-Pseudo-Labeling algorithm. □

To prove Theorem 12, we use the following Lemma:

**Lemma 19.** *Assume that $\mathcal{A}$ is a **teacher** for some distribution $\mathcal{D}$, with sample complexity $\widetilde{m}$. Then, for every $\epsilon, \delta \in (0,1)$, taking $m \geq \widetilde{m}\left(\frac{\epsilon}{3}, \frac{\gamma(\mathcal{D})}{2}\right)$ and $k \geq \frac{64}{\gamma(\mathcal{D})^2} \log\left(\frac{3}{\epsilon\delta}\right)$ we get that w.p. at least $1 - \delta$ over the choice of $S_1,\ldots,S_k$, it holds that:*
$$\mathbb{P}_{\mathbf{x} \sim \mathcal{D}}\left[f_{\mathcal{D}}^*(\mathbf{x})\frac{1}{k}\sum_{i=1}^{k}\mathcal{A}(S_i)(\mathbf{x}) \leq \gamma(\mathcal{D})/4\right] \leq \epsilon$$

*Proof.* of Lemma 19. Let $\mathcal{X}' \subseteq \mathcal{X}$ be the subset of points $\mathbf{x} \in \mathcal{X}$ satisfying the assumptions of Lemma 18 with $\gamma = \frac{\gamma(\mathcal{D})}{2}$ and $\tau = \frac{\gamma(\mathcal{D})}{4}$. Observe that, using the union bound, and the properties of the teacher $\mathcal{A}$:

$$\mathbb{P}_{\mathbf{x} \sim \mathcal{D}}\left[\mathbf{x} \notin \mathcal{X}'\right]$$
$$\leq \mathbb{P}_{\mathbf{x} \sim \mathcal{D}}\left[\mathbb{P}_{\mathcal{A}(\mathcal{D}^m)}\left[f_{\mathcal{A}(\mathcal{D}^m)}^*(\mathbf{x}) \mid \mathbf{x}\right] > \mathbb{P}_{\mathcal{A}(\mathcal{D}^m)}\left[-f_{\mathcal{A}(\mathcal{D}^m)}^*(\mathbf{x}) \mid \mathbf{x}\right] + \gamma\right]$$
$$+ \mathbb{P}_{\mathbf{x} \sim \mathcal{D}}\left[f_{\mathcal{A}(\mathcal{D}^m)}^*(\mathbf{x}) \neq f_{\mathcal{D}}^*(\mathbf{x})\right]$$
$$\leq \epsilon/3 + L_{\mathcal{D}^*}\left(f_{\mathcal{A}(\mathcal{D}^m)}^*\right) \leq \frac{2\epsilon}{3}$$

Let $\delta' = \frac{\epsilon\delta}{3}$. Fix some $\mathbf{x} \in \mathcal{X}'$, and from Lemma 18 we have:

$$\mathop{\mathbb{E}}_{S_1,\ldots,S_k \sim \mathcal{D}^m} \mathbf{1}\{f_{\mathcal{D}}^*(\mathbf{x})\frac{1}{k}\sum_i \mathcal{A}_{\mathrm{ens}}(S_i)(\mathbf{x}) \leq \tau\} \leq \exp\left(-\frac{k(\gamma-\tau)^2}{4}\right) \leq \delta'$$

Therefore, we get:

$$\mathop{\mathbb{E}}_{S_1,\ldots,S_k \sim \mathcal{D}^m} \mathbb{P}_{\mathbf{x} \sim \mathcal{D}}\left[f_{\mathcal{D}}^*(\mathbf{x})\frac{1}{k}\sum_i \mathcal{A}_{\mathrm{ens}}(S_i)(\mathbf{x}) \leq \tau \mid \mathbf{x} \in \mathcal{X}'\right]$$
$$= \mathop{\mathbb{E}}_{\mathbf{x}}\left[\mathop{\mathbb{E}}_{S_1,\ldots,S_k \sim \mathcal{D}^m} \mathbf{1}\{f_{\mathcal{D}}^*(\mathbf{x})\frac{1}{k}\sum_i \mathcal{A}_{\mathrm{ens}}(S_i)(\mathbf{x}) \leq \tau\} \mid \mathbf{x} \in \mathcal{X}'\right] \leq \delta'$$

Using Markov's inequality we get that w.p. at least $1 - \frac{3\delta'}{\epsilon}$ we have

$$\mathbb{P}_{\mathbf{x} \sim \mathcal{D}}\left[f_{\mathcal{D}}^*(\mathbf{x})\frac{1}{k}\sum_i \mathcal{A}_{\mathrm{ens}}(S_i)(\mathbf{x}) \leq \tau \mid \mathbf{x} \in \mathcal{X}'\right] \leq \frac{\epsilon}{3}$$

and in this case we have

$$\mathbb{P}_{\mathbf{x} \sim \mathcal{D}}\left[f_{\mathcal{D}}^*(\mathbf{x})\frac{1}{k}\sum_i \mathcal{A}_{\mathrm{ens}}(S_i)(\mathbf{x}) \leq \tau\right]$$
$$\leq \mathbb{P}_{\mathbf{x} \sim \mathcal{D}}\left[f_{\mathcal{D}}^*(\mathbf{x})\frac{1}{k}\sum_i \mathcal{A}_{\mathrm{ens}}(S_i)(\mathbf{x}) \leq \tau \mid \mathbf{x} \in \mathcal{X}'\right] + \mathbb{P}_{\mathbf{x} \sim \mathcal{D}}[\mathbf{x} \notin \mathcal{X}'] \leq \epsilon$$

□

*Proof.* of Theorem 12. Fix $\epsilon > 0$ and let $\epsilon' = \frac{\gamma(\mathcal{D})\epsilon}{18}$. Fix a sequence of $k$ subsets of examples $\mathcal{S} = (S_1, \ldots, S_k)$, and let $\widetilde{\mathcal{D}}_\mathcal{S}$ be the distribution over $\mathcal{X} \times \mathcal{Y}$ given by sampling $\mathbf{x} \sim \mathcal{D}$, sampling $i \sim \{1, \ldots, k\}$ and returning $(\mathbf{x}, y)$ where $y = \mathcal{A}(S_i)(\mathbf{x})$. Let $\widetilde{S}_\mathcal{S}$ be an i.i.d. sample of size $m'$ from $\widetilde{\mathcal{D}}_\mathcal{S}$. Let $h_\mathcal{S} = \mathrm{ERM}_\mathcal{H}(\widetilde{S}_\mathcal{S})$. By the Fundamental Theorem of Statistical Learning (e.g. Theorem 6.8 in [25]) w.p. at least $1 - \epsilon'$ over sampling $\widetilde{S}_\mathcal{S}$ we have:

$$
\begin{aligned}
L_{\widetilde{\mathcal{D}}_\mathcal{S}}(h_\mathcal{S}) &\le \inf_{h \in \mathcal{H}} L_{\widetilde{\mathcal{D}}_\mathcal{S}}(h) + \epsilon' \le L_{\widetilde{\mathcal{D}}_\mathcal{S}}(f_\mathcal{D}^*) + \epsilon' \\
&= \mathop{\mathbb{E}}_{x \sim \mathcal{D}}[\mathbf{1}\{f_\mathcal{D}^*(\mathbf{x}) \ne y\}] \le \mathop{\mathbb{E}}_{x \sim \mathcal{D}}[\mathbf{1}\{f_\mathcal{D}^*(\mathbf{x}) \ne \mathcal{A}_{\mathrm{ens}}(\mathcal{S})(\mathbf{x})\} + \mathbf{1}\{\mathcal{A}_{\mathrm{ens}}(\mathcal{S})(\mathbf{x}) \ne y\}] + \epsilon' \\
&= L_{\mathcal{D}^*}(\mathcal{A}_{\mathrm{ens}}(\mathcal{S})) + L_{\widetilde{\mathcal{D}}_\mathcal{S}}(\mathcal{A}_{\mathrm{ens}}(\mathcal{S})) + \epsilon'
\end{aligned}
$$

**Claim**: If $\mathcal{S}$ satisfies $\gamma_{\epsilon'}(\widetilde{\mathcal{D}}_\mathcal{S}) > 0$ then w.p. at least $1 - \epsilon'$ over the choice of $\widetilde{S}_\mathcal{S} \sim \widetilde{\mathcal{D}}_\mathcal{S}^{m'}$

$$
L_{\mathcal{D}^*}(h_\mathcal{S}) \le (L_{\mathcal{D}^*}(\mathcal{A}_{\mathrm{ens}}(\mathcal{S})) + \epsilon')\left(1 + \gamma_{\epsilon'}(\widetilde{\mathcal{D}}_\mathcal{S})^{-1}\right)
$$

**Proof**: W.p. at least $1 - \epsilon'$ we have $L_{\widetilde{\mathcal{D}}_\mathcal{S}}(h_\mathcal{S}) \le L_{\widetilde{\mathcal{D}}_\mathcal{S}}(\mathcal{A}_{\mathrm{ens}}(\mathcal{S})) + L_{\mathcal{D}^*}(\mathcal{A}_{\mathrm{ens}}(\mathcal{S})) + \epsilon'$. Notice that by definition of $\widetilde{\mathcal{D}}_\mathcal{S}$, we have that $\mathcal{A}_{\mathrm{ens}}(\mathcal{S})$ is the Bayes optimal classifier for $\widetilde{\mathcal{D}}_\mathcal{S}$. Therefore, by Lemma 3 we have $L_{\widetilde{\mathcal{D}}_\mathcal{S}^*}(h_\mathcal{S}) \le \frac{\epsilon' + L_{\mathcal{D}^*}(\mathcal{A}_{\mathrm{ens}}(\mathcal{S}))}{\gamma_{\epsilon'}(\widetilde{\mathcal{D}}_\mathcal{S})} + \epsilon'$. Now, we have:

$$
\begin{aligned}
L_{\mathcal{D}^*}(h_\mathcal{S}) &= \mathop{\mathbb{E}}_{\mathbf{x} \sim \mathcal{D}}[\mathbf{1}\{h_\mathcal{S}(\mathbf{x}) \ne f_\mathcal{D}^*(\mathbf{x})\}] \\
&\le \mathop{\mathbb{E}}_{\mathbf{x} \sim \mathcal{D}}[\mathbf{1}\{h_\mathcal{S}(\mathbf{x}) \ne \mathcal{A}_{\mathrm{ens}}(\mathcal{S})(\mathbf{x})\} + \mathbf{1}\{\mathcal{A}_{\mathrm{ens}}(\mathcal{S})(\mathbf{x}) \ne f_\mathcal{D}^*(\mathbf{x})\}] \\
&= L_{\widetilde{\mathcal{D}}_\mathcal{S}^*}(h_\mathcal{S}) + L_{\mathcal{D}^*}(\mathcal{A}_{\mathrm{ens}}(\mathcal{S})) \le \frac{\epsilon' + L_{\mathcal{D}^*}(\mathcal{A}_{\mathrm{ens}}(\mathcal{S}))}{\gamma_{\epsilon'}(\widetilde{\mathcal{D}}_\mathcal{S})} + \epsilon' + L_{\mathcal{D}^*}(\mathcal{A}_{\mathrm{ens}}(\mathcal{S}))
\end{aligned}
$$

**Claim**: W.p. $> 1 - \epsilon'$ over the choice of $\mathcal{S}$, we have $\gamma_{\epsilon'}(\widetilde{\mathcal{D}}_\mathcal{S}) \ge \frac{\gamma(\mathcal{D})}{4}$ and $L_{\mathcal{D}^*}(\mathcal{A}_{\mathrm{ens}}(\mathcal{S})) \le \epsilon'$.

**Proof**: By Lemma 19, since $m \ge \widetilde{m}\left(\frac{\epsilon'}{3}, \frac{\gamma(\mathcal{D})}{2}, \frac{\epsilon'}{3}\right)$ and $k \ge \frac{64}{\gamma(\mathcal{D})^2} \log\left(\frac{3}{(\epsilon')^2}\right)$ we have, w.p. $> 1 - \epsilon'$ over the choice of $\mathcal{S}$, that

$$
\begin{aligned}
&\mathop{\mathbb{P}}_{\mathbf{x} \sim \mathcal{D}}\left[\left(\mathop{\mathbb{P}}_{i \sim [k]}[\mathcal{A}(S_i)(\mathbf{x}) = f_\mathcal{D}^*(\mathbf{x})|\mathbf{x}] - \mathop{\mathbb{P}}_{i \sim [k]}[\mathcal{A}(S_i)(\mathbf{x}) = -f_\mathcal{D}^*(\mathbf{x})|\mathbf{x}]\right) > \gamma(\mathcal{D})/4\right] \\
&= \mathop{\mathbb{P}}_{\mathbf{x} \sim \mathcal{D}}\left[f_\mathcal{D}^*(\mathbf{x})\frac{1}{k}\sum_i \mathcal{A}(S_i)(\mathbf{x}) > \gamma(\mathcal{D})/4\right] \le \epsilon'
\end{aligned}
$$

which immediately implies the required.

From the above two claims, w.p. at least $1 - 2\epsilon'$ over the choice of $\mathcal{S}, \widetilde{S}_\mathcal{S}$ we have

$$
L_{\mathcal{D}^*}(h_\mathcal{S}) \le 2\epsilon'\left(1 + \gamma_{\epsilon'}(\widetilde{\mathcal{D}}_\mathcal{S})^{-1}\right) \le \frac{16\epsilon'}{\gamma(\mathcal{D})}
$$

and therefore $\mathbb{E}_{\mathcal{S}, \widetilde{S}_\mathcal{S}} L_{\mathcal{D}^*}(h_\mathcal{S}) \le \frac{16\epsilon'}{\gamma(\mathcal{D})} + 2\epsilon' \le \frac{18\epsilon'}{\gamma(\mathcal{D})} = \epsilon$. □

## C    Section 4 Additional Details and proofs

### C.1    Well-Clustered Data and Lipschitz Classes

We now show that under certain clustering assumptions, many learning methods can be teachers. First, we study a simplified case of a distribution supported on a finite set. The following theorem shows that when the hypothesis class shatters the support of the distribution, $\mathrm{ERM}_\mathcal{H}$ is a teacher with sample complexity $\tilde{O}(k/\epsilon)$.

**Theorem 20.** *Fix some hypothesis class $\mathcal{H}$, and let $\mathcal{D}$ be some distribution over $\mathcal{X} \times \mathcal{Y}$ such that $|\mathrm{supp}(\mathcal{D}_\mathcal{X})| = k \le \mathrm{VC}(\mathcal{H})$ and the support of $\mathcal{D}_\mathcal{X}$ is shattered. Then, $\mathrm{ERM}_\mathcal{H}$ is a teacher, with sample complexity $\widetilde{m}(\epsilon, \tau) = \frac{2k \log(2k/\varepsilon)}{\varepsilon}$.*

Contrast this with Theorem 2, where we show that when $\text{VC}(\mathcal{H}) = d$, ERM is a learner with sample complexity $m(\varepsilon) = \tilde{O}\left(\frac{\text{VC}(\mathcal{H})}{\epsilon^2 \gamma(\mathcal{D})^2}\right)$. This shows that sampling can be achieved in this case without a dependence on $1/\gamma^2$, as would be needed in order to get a learner. In fact, Theorem 6 shows that the dependence on $1/\gamma^2$ in the sample complexity of a learner cannot be avoided.

We proceed to discuss a more general version of Theorem 20 where $\mathcal{D}$ is well-clustered in $k$ balls of small radius (similar to a Mixture of Gaussians with low variance). In this case, we study $L$-Lipschitz hypothesis classes, defined as follows:

**Definition 21.** *A hypothesis class $\mathcal{H}$ is $L$-Lipschitz if for every $h \in \mathcal{H}$ there exists some $\hat{h} : \mathcal{X} \to \mathbb{R}$ such that $\hat{h}$ is $L$-Lipschitz and $h(\mathbf{x}) = \text{sign}\,\hat{h}(\mathbf{x})$ for all $\mathbf{x} \in \mathcal{X}$.*

We note that a large family of learning methods such as bounded norm linear classifiers, kernel machines and shallow neural networks with Lipschitz activations (e.g., ReLU) are Lipschitz classes. For learning $L$-Lipschitz classes, we study the ERM rule with respect to the hinge-loss (over the real-valued output) instead of the zero-one loss. Namely, we define $\text{ERM}_{\mathcal{H}}^{hinge}(S) = \arg\min_{h \in \mathcal{H}} \mathbb{E}_{(\mathbf{x},y)}\left[\ell_{\text{hinge}}\left(y, \hat{h}(\mathbf{x})\right)\right]$, where $\ell_{\text{hinge}}(y, \hat{y}) = \max(1 - y\hat{y}, 0)$.

We use the hinge-loss as it is often required that the output of a real-valued hypothesis separates the data with some margin. Indeed, since the zero-one loss is invariant to scale, the $L$-Lipschitz assumption under the zero-one loss is meaningless, since the hypothesis can always be scaled down to satisfy any Lipschitz bound. So, when the data is well-clustered and the hypothesis class $\mathcal{H}$ is $L$-Lipschitz, $\text{ERM}_{\mathcal{H}}^{hinge}$ is a teacher with sample complexity $\tilde{O}(\frac{k}{\gamma^2 \varepsilon})$. While this bound does depend on $1/\gamma^2$, it still improves the sample complexity of learning derived from Theorem 2.

**Theorem 22.** *For $L$-Lipschitz class $\mathcal{H}$, and some $\lambda$-Lipschitz distribution $\mathcal{D}$ s.t. $\text{supp}(\mathcal{D}_{\mathcal{X}}) \subseteq \cup_{i=1}^{k} B(\mathbf{c}_i, r)$, where $r = \frac{\gamma}{2 \max(\lambda, 3L)}$ and $k \leq \text{VC}(\mathcal{H})$ so the set of balls $B(\mathbf{c}_i, r)$ can be shattered. Then, $\text{ERM}_{\mathcal{H}}^{hinge}$ is a teacher, with sample complexity $\widetilde{m}(\epsilon, \tau) = \tilde{O}(\frac{k \log(2k/\varepsilon)}{\gamma^2 \varepsilon})$.*

## C.2 Proofs for Section 4

Using standard measure-theoretic arguments, we show that for any distribution $\mathcal{D}$, such a cover exists:

**Lemma 23.** *For a every distribution $\mathcal{D}$ over $\mathcal{X} \times \mathcal{Y}$ there exists a function $m_c : (0, 1) \times (0, 1) \to \mathbb{N}$ s.t. for every $\epsilon, \delta \in (0, 1)$ there exists a subset $\mathcal{X}' \subseteq \mathcal{X}$ satisfying:*

- $\mathbb{P}_{\mathbf{x} \sim \mathcal{D}_{\mathcal{X}}}[\mathbf{x} \notin \mathcal{X}'] \leq \delta$
- *If $m \geq m_c(\epsilon, \delta)$, for all $\mathbf{x} \in \mathcal{X}'$ it holds that $\mathbb{P}_{S \sim \mathcal{D}_{\mathcal{X}}^m}[d(\mathbf{x}, S) > \epsilon] \leq \delta$.*

*Proof.* of Lemma 23 Fix some $\epsilon, \delta \in (0, 1)$ and let $\delta' = \delta/2, \epsilon' = \epsilon/2$. For some $\mathbf{x}_0 \in \mathcal{X}$ and let $B_r(\mathbf{x}_0)$ be the closed ball of radius $r$ around $\mathbf{x}_0$, i.e.

$$B_r(\mathbf{x}_0) = \{\mathbf{x} \in \mathcal{X} \ : \ d(\mathbf{x}_0, \mathbf{x}) \leq r\}$$

Now, for some $\mathbf{x}_0 \in \mathcal{X}$, observe that $\mathcal{X} = \cup_{r=1}^{\infty} B_r(\mathbf{x}_0)$, and therefore we have:

$$1 = \mathcal{D}_{\mathcal{X}}(\mathcal{X}) = \mathcal{D}_{\mathcal{X}}(\cup_{r=1}^{\infty} B_r(\mathbf{x}_0)) = \lim_{r \to \infty} \mathcal{D}_{\mathcal{X}}(B_r(\mathbf{x}_0))$$

So, there exists some $r$ s.t. $\mathcal{D}_{\mathcal{X}}(B_r(\mathbf{x}_0)) \geq 1 - \delta'$. Now, since $B_r(\mathbf{x}_0)$ is closed and bounded in $(\mathcal{X}, d)$, from the Heine-Borel property we get that $B_r(\mathbf{x}_0)$ is also compact. Since $B_r(\mathbf{x}_0) \subseteq \cup_{\mathbf{x} \in B_r(\mathbf{x}_0)} B_{\epsilon'}(\mathbf{x})$, there exists some finite subset $C \subseteq B_r(\mathbf{x}_0)$ such that $B_r(\mathbf{x}_0) \subseteq \cup_{\mathbf{x} \in C} B_{\epsilon'}(\mathbf{x})$. Now, let $C' \subseteq C$ be the subset of balls that have at least $\delta'/|C|$ mass under $\mathcal{D}_{\mathcal{X}}$, namely:

$$C' = \left\{\mathbf{x} \in C \ : \ \mathcal{D}_{\mathcal{X}}(B_{\epsilon'}(\mathbf{x})) \geq \frac{\delta'}{|C|}\right\}$$

Let $m = \left\lceil \frac{|C|}{\delta'} \log\left(\frac{|C|}{\delta'}\right)\right\rceil$, and observe that for every $\mathbf{x} \in C'$ we have:

$$\mathbb{P}_{S \sim \mathcal{D}_{\mathcal{X}}^m}[S \cap B_{\epsilon'}(\mathbf{x}) = \emptyset] = \mathbb{P}_{\mathbf{x}' \sim \mathcal{D}_{\mathcal{X}}}[\mathbf{x}' \notin B_{\epsilon'}(\mathbf{x})]^m \leq \left(1 - \frac{\delta'}{|C|}\right)^m \leq \exp\left(-\frac{m\delta'}{|C|}\right) \leq \frac{\delta'}{|C|}$$

Using the union bound, w.p. at least $1 - \delta'$ it holds that for all $\mathbf{x} \in C'$ ther exists $\mathbf{x}' \in S$ s.t. $\mathbf{x}' \in B_{\epsilon'}(\mathbf{x})$. Denote by $\mathcal{X}'$ all the points in $\mathcal{X}$ that are covered by $C'$, namely $\mathcal{X}' = \cup_{\mathbf{x} \in C'} B_{\epsilon'}(\mathbf{x})$.

**Claim**: $\mathcal{X} \setminus \mathcal{X}' \subseteq (\mathcal{X} \setminus B_r(\mathbf{x}_0)) \cup (\cup_{\mathbf{x} \in C \setminus C'} B_{\epsilon'}(\mathbf{x}))$

**Proof**: Let $\mathbf{x} \in \mathcal{X} \setminus \mathcal{X}'$ and we need to show $\mathbf{x} \in (\mathcal{X} \setminus B_r(\mathbf{x}_0)) \cup (\cup_{\mathbf{x} \in C \setminus C'} B_{\epsilon'}(\mathbf{x}))$. If $\mathbf{x} \notin B_r(\mathbf{x}_0)$ we are done. Otherwise, if $\mathbf{x} \in B_r(\mathbf{x}_0)$, since $B_r(\mathbf{x}_0) \subseteq \cup_{\mathbf{x}' \in C} B_{\epsilon'}(\mathbf{x})$ there exists some $\mathbf{x}' \in C$ s.t. $\mathbf{x} \in B_{\epsilon'}(\mathbf{x}')$, and $\mathbf{x}' \notin C'$ since otherwise we would have $\mathbf{x} \in \mathcal{X}'$.

**Claim**: $\mathbb{P}_{\mathbf{x} \sim \mathcal{D}_{\mathcal{X}}} [\mathbf{x} \notin \mathcal{X}'] \leq 2\delta'$

**Proof**: Using the union bound and the previous result:

$$\mathbb{P}_{\mathbf{x} \sim \mathcal{D}_{\mathcal{X}}} [\mathbf{x} \notin \mathcal{X}'] \leq \mathbb{P}_{\mathbf{x} \sim \mathcal{D}_{\mathcal{X}}} [\mathbf{x} \notin B_r(\mathbf{x}_0)] + \sum_{\mathbf{x}' \in C \setminus C'} \mathbb{P}_{\mathbf{x} \sim \mathcal{D}_{\mathcal{X}}} [\mathbf{x} \in B_{\epsilon'}(\mathbf{x}')]$$

$$\leq \delta' + |C \setminus C'| \frac{\delta'}{|C|} \leq 2\delta'$$

**Claim**: W.p. at least $1 - \delta'$ over the choice of $S \sim \mathcal{D}_{\mathcal{X}}^m$, for all $\mathbf{x} \in \mathcal{X}'$ it holds that $d(\mathbf{x}, S) \leq \epsilon$.

**Proof**: From what we showed, w.p. at least $1 - \delta'$, for all $\mathbf{x} \in C'$ there exits $\mathbf{x}' \in S$ s.t. $\mathbf{x}' \in B_{\epsilon'}(\mathbf{x})$. Assume this holds, and let $\mathbf{x} \in \mathcal{X}'$. By definition of $\mathcal{X}'$ there exists some $\hat{\mathbf{x}} \in C'$ s.t. $d(\mathbf{x}, \hat{\mathbf{x}}) \leq \epsilon'$. So, there is some $\mathbf{x}' \in S$ s.t. $d(\mathbf{x}', \hat{\mathbf{x}}) \leq \epsilon'$, and therefore $d(\mathbf{x}, \mathbf{x}') \leq 2\epsilon' = \epsilon$ and we get the required.

Now, the required follows from the last two claims. $\qquad\qquad\square$

*Proof.* of Theorem 13.

Let $\mathcal{D}$ be some $\lambda$-Lipschitz distribution. Let $m_c(\cdot, \cdot)$ be a function satisfying the conditions guaranteed by Lemma 23 for the distribution $\mathcal{D}$. Then, we prove that the $\mathcal{A}_{1\text{-NN}}$ is a **sampler** for $\mathcal{D}$, with distributional sample complexity $\widetilde{m}(\epsilon) = m_c\left(\frac{\epsilon}{2\lambda}, \frac{\epsilon}{12}\right)$.

Fix $\epsilon \in (0, 1)$ and let $\epsilon' = \frac{\epsilon}{2\lambda}, \delta' = \frac{\epsilon}{12}$. Let $m_c$ be the function guaranteed by Lemma 23, and let $\mathcal{X}'$ be the subset guaranteed by the same Theorem (given the choice of $\epsilon', \delta'$). Fix some $\mathbf{x} \in \mathcal{X}'$. Denote $q := \mathbb{P}_{S \sim \mathcal{D}^m} [d(\mathbf{x}, S) \leq \epsilon']$ (the probability to get a good cover). By Lemma 23, for $m = m_c(\epsilon', \delta')$ we get that $q \geq 1 - \delta'$. For every $y \in \mathcal{Y}$, denote $p_{\mathbf{x}}(y) := \mathbb{P}_{\mathcal{D}}[y|\mathbf{x}]$, and we have:

$$\left| \mathbb{P}_{S \sim \mathcal{D}^m} [\mathcal{A}_{\text{NN}}(S)(\mathbf{x}) = y] - p_{\mathbf{x}}(y) \right| \leq q \left| \mathbb{P}_{S \sim \mathcal{D}^m} [\mathcal{A}_{\text{NN}}(S)(\mathbf{x}) = y | d(\mathbf{x}, S) \leq \epsilon] - p_{\mathbf{x}}(y) \right|$$

$$+ (1 - q) \left| \mathbb{P}_{S \sim \mathcal{D}^m} [\mathcal{A}_{\text{NN}}(S)(\mathbf{x}) = y | d(\mathbf{x}, S) > \epsilon] - p_{\mathbf{x}}(y) \right|$$

$$\leq \left| \mathbb{P}_{\mathcal{D}}[y | \pi(\mathbf{x}, S), d(\mathbf{x}, S) \leq \epsilon] - p_{\mathbf{x}}(y) \right| + 2\delta' \leq \lambda\epsilon' + 2\delta'$$

From the above we get:

$$\mathbb{E}_{\mathbf{x}} \sum_y \left| \mathbb{P}_{S \sim \mathcal{D}^m} [\mathcal{A}_{\text{NN}}(S)(\mathbf{x}) | \mathbf{x}] - \mathbb{P}[y | \mathbf{x}] \right|$$

$$\leq \mathbb{E}_{\mathbf{x} | \mathbf{x} \in \mathcal{X}'} \sum_y \left| \mathbb{P}_{S \sim \mathcal{D}^m} [\mathcal{A}_{\text{NN}}(S)(\mathbf{x}) | \mathbf{x}] - \mathbb{P}[y | \mathbf{x}] \right| + 2 |\mathcal{Y}| \mathbb{P}_{\mathbf{x} \sim \mathcal{D}} [\mathbf{x} \notin \mathcal{X}]$$

$$\leq \lambda\epsilon' + 6\delta' \leq \epsilon$$

and therefore the required follows. $\qquad\qquad\square$

*Proof.* of Theorem 14

Let $\mathcal{D}$ be some $\lambda$-Lipschitz distribution. Let $m_c(\cdot, \cdot)$ be a function satisfying the conditions guaranteed by Lemma 23 for the distribution $\mathcal{D}$. Then, we prove that the $\mathcal{A}_{k\text{-NN}}$ algorithm is a **teacher** for $\mathcal{D}$, with sample complexity $\widetilde{m}(\epsilon, \tau) = k \cdot m_c\left(\frac{\tau}{4\lambda}, \min\left\{\epsilon, \frac{\tau}{4k}\right\}\right)$.

Fix $\epsilon \in (0, 1), \tau \in (0, \gamma(\mathcal{D}))$ and let $\epsilon' = \frac{\tau}{4\lambda}, \delta' = \min\left\{\epsilon, \frac{\tau}{4k}\right\}$. Let $m_c$ be the function guaranteed by Lemma 23, and let $\mathcal{X}'$ be the subset guaranteed by the same Theorem (given the choice of $\epsilon', \delta'$). Fix some $\mathbf{x} \in \mathcal{X}'$. Let $\mathcal{S}$ be the set of subsets of $\mathcal{X}$ such that $S_{\mathcal{X}} \in \mathcal{S}$ if and only if for all $\mathbf{x}' \in k\text{-}\pi(\mathbf{x}, S_{\mathcal{X}})$ it holds that $d(\mathbf{x}, \mathbf{x}') \leq \epsilon'$. Let $m = k \cdot m_c(\epsilon', \delta')$.

**Claim:** $\mathbb{P}_{S\sim\mathcal{D}^m}\left[S_{\mathcal{X}}\in\mathcal{S}\right]\geq 1-k\delta'$

**Proof:** For every set $S\subseteq\mathcal{X}\times\mathcal{Y}$ of size $m$, split $S$ to blocks of $k$ examples $S^{(1)},\ldots,S^{(k)}$ each of size $m_c(\epsilon',\delta')$. By Lemma 23, for every $i$ it holds that $\mathbb{P}_{S^{(i)}\sim\mathcal{D}^{m/k}}\left[d\left(\mathbf{x},S_{\mathcal{X}}^{(i)}\right)>\epsilon'\right]\leq\delta'$. Using the union bound, with probability at least $1-k\delta'$ if holds that for every $i\in[k]$ we have $d\left(\mathbf{x},S_{\mathcal{X}}^{(i)}\right)\leq\epsilon'$, in which case there are at least $k$ examples in $S$ with distance $\leq\epsilon'$ to $\mathbf{x}$, so $S_{\mathcal{X}}\in\mathcal{S}$.

**Claim:** $\mathbb{P}_{S\sim\mathcal{D}^m}\left[\mathcal{A}_{\text{k-NN}}(S)(\mathbf{x})=f_{\mathcal{D}}^*(\mathbf{x})|S_{\mathcal{X}}\in\mathcal{S}\right]\geq\frac{1}{2}+\frac{\gamma(\mathcal{D})}{2}-\frac{\tau}{4}$

**Proof:** Fix some $S_{\mathcal{X}}\in\mathcal{S}$, and w.l.o.g. assume that k-$\pi(\mathbf{x},S_{\mathcal{X}})=\{\mathbf{x}_1,\mathbf{x}_2,\ldots,\mathbf{x}_k\}$. Then,

$$\mathbb{P}_{S'\sim\mathcal{D}^m}\left[\mathcal{A}_{\text{k-NN}}(S)(\mathbf{x})=f_{\mathcal{D}}^*(\mathbf{x})\mid S'_{\mathcal{X}}=S_{\mathcal{X}}\right]$$

$$=\mathbb{P}_{S'\sim\mathcal{D}^m}\left[\text{sign}\left(\sum_{i=1}^{k}y_i\right)=f_{\mathcal{D}}^*(\mathbf{x})\mid S'_{\mathcal{X}}=S_{\mathcal{X}}\right]$$

Denote $p_i=\mathbb{P}_{S'\sim\mathcal{D}^m}\left[y_i=f_{\mathcal{D}}^*(\mathbf{x})|S'_{\mathcal{X}}=S_{\mathcal{X}}\right]=\mathbb{P}_{\mathcal{D}}[f_{\mathcal{D}}^*(\mathbf{x})|\mathbf{x}_i]$. Now, observe that:

$$p_i\geq\mathbb{P}_{\mathcal{D}}[f_{\mathcal{D}}^*(\mathbf{x})|\mathbf{x}]-\lambda d(\mathbf{x},\mathbf{x}_i)\geq\mathbb{P}_{\mathcal{D}}[f^*(\mathbf{x})|\mathbf{x}]-\lambda\epsilon'\geq\frac{1}{2}+\frac{\gamma(\mathcal{D})}{2}-\lambda\epsilon'\geq\frac{1}{2}+\frac{\gamma(\mathcal{D})}{2}-\frac{\tau}{4}$$

where the first inequality uses the $\lambda$-Lipschitz property of $\mathcal{D}$, and the third inequality is by definition of $\gamma(\mathcal{D})$. Now, from the Conodorcet Jury Theorem in [3], it holds that:

$$\mathbb{P}_{S'\sim\mathcal{D}^m}\left[\text{sign}\left(\sum_{i=1}^{k}y_i\right)=f_{\mathcal{D}}^*(\mathbf{x})\mid S'_{\mathcal{X}}=S_{\mathcal{X}}\right]\geq\frac{1}{k}\sum_{i=1}^{k}p_i\geq\frac{1}{2}+\frac{\gamma(\mathcal{D})}{2}-\frac{\tau}{4}$$

and the claim follows from the law of total probability.

**Claim:** For every $\mathbf{x}\in\mathcal{X}'$ it holds that $\mathbb{P}_{S\sim\mathcal{D}^m}\left[\mathcal{A}_{\text{k-NN}}(S)(\mathbf{x})=f_{\mathcal{D}}^*(\mathbf{x})\right]\geq\frac{1}{2}+\frac{\gamma(\mathcal{D})-\tau}{2}$.

**Proof:** Observe that, using the previous claims:

$$\mathbb{P}_{S\sim\mathcal{D}^m}\left[\mathcal{A}_{\text{k-NN}}(S)(\mathbf{x})\neq f_{\mathcal{D}}^*(\mathbf{x})\right]\leq\mathbb{P}_{S\sim\mathcal{D}^m}\left[\mathcal{A}_{\text{k-NN}}(S)(\mathbf{x})\neq f_{\mathcal{D}}^*(\mathbf{x})|S_{\mathcal{X}}\in\mathcal{S}\right]+\mathbb{P}_{S\sim\mathcal{D}^m}\left[S\notin\mathcal{S}\right]$$

$$<\frac{1}{2}-\frac{\gamma(\mathcal{D})}{2}+\frac{\tau}{4}+k\delta'\leq\frac{1}{2}-\frac{\gamma(\mathcal{D})-\tau}{2}$$

By the previous claim, it follows that for all $\mathbf{x}\in\mathcal{X}'$ we have $f_{\mathcal{A}_{\text{k-NN}}(\mathcal{D}^m)}^*(\mathbf{x})=f_{\mathcal{D}}^*(\mathbf{x})$, and using the fact that $\mathbb{P}_{\mathbf{x}\sim\mathcal{D}}\left[\mathbf{x}\notin\mathcal{X}'\right]\leq\delta'\leq\epsilon$ the first condition for teacher holds. Since we also have $\mathbb{P}_{\mathbf{x}\sim\mathcal{D}}\left[\mathbf{x}\notin\mathcal{X}'\right]\leq\delta'\leq\delta$, by the previous claim we get that $\gamma_\delta(\mathcal{A}_{\text{k-NN}}(\mathcal{D}^m))\geq\gamma(\mathcal{D})-\tau$, and the second condition in the definition of teacher holds. $\square$

*Proof.* of Theorem 15.

In the one-dimensional case, i.e. when $\mathcal{X}=\mathbb{R}$, Theorem 3.3 from [24] shows that $R(S)$ gives the linear spline interpolation of the data points. Namely, let $\hat{\theta}:=R(S)$, and assume that $S=\{(x_1,y_1),\ldots,(x_m,y_m)\}$ is sorted such that $x_1<x_2<\cdots<x_m$ (assuming there are no repeated samples). Then, for every $i\in[m]$ and for all $x\in[x_i,x_{i+1}]$ it holds that

$$h_{\hat{\theta}}(x)=y_i+\frac{y_{i+1}-y_i}{x_{i+1}-x_i}(x-x_i)$$

In this case, it can be easily shown that for all $x\in[x_1,x_m]$ we have $\text{sign}\,h_{\hat{\theta}}(x)=1\text{-NN}(S)(x)$, so training a network with bounded-norm weights (and unbounded width) behaves like nearest neighbour classification over the range covered by the sample. Using this, we show that ReLU networks in this setting are samplers.

Let $\epsilon>0$ and let $\varepsilon'=\varepsilon/4$. We begin with the following claim:

**Claim**: There exist numbers $a<b$ such that $\mathbb{P}_{x\sim\mathcal{D}}\left[x\leq a\right]=\mathbb{P}_{x\sim\mathcal{D}}\left[x\geq b\right]=\epsilon'$.

**Proof**: By assumption the function $F(a)=\mathbb{P}[x\leq a]$ is continuous and $\lim_{a\to\infty}=1$, $\lim_{a\to-\infty}=0$ thus by the intermediate value theorem we have there exist $a,b$ such that $F(a)=\mathbb{P}[x\leq a]=\varepsilon'$ and $F(b)=\mathbb{P}[x\leq b]=1-\varepsilon'$.

Assume we sample $S \sim \mathcal{D}^m$, and sort it s.t. $S = ((x_1, y_1), \ldots, (x_m, y_m))$ where $x_1 < x_2 < \cdots < x_m$.

**Claim**: Fix $\delta' > 0$, and assume that $m \geq \frac{\log(2/\delta')}{\epsilon'}$. Then, w.p. at least $1 - \delta'$ over the choice of $S$, it holds that

$$\mathbb{P}_{x \sim \mathcal{D}} \left[ x \notin [x_1, x_m] \right] \leq 2\epsilon'$$

**Proof**: Let $a, b$ be the numbers guaranteed by the previous claim. Then, we have

$$\mathbb{P}_{S \sim \mathcal{D}^m} \left[ x_1 > a \right] = \mathbb{P}_{S \sim \mathcal{D}^m} \left[ \forall (x, y) \in S, \ x > a \right] = (1 - \epsilon')^m \leq e^{-\epsilon' m} \leq \frac{\delta'}{2}$$

and similarly we get $\mathbb{P}_{S \sim \mathcal{D}^m} \left[ x_m < b \right] \leq \frac{\delta'}{2}$. So, from the union bound, w.p. at least $1 - \delta'$ it holds that $x_1 \leq a$ and $x_m \geq b$. In this case, we have:

$$\mathbb{P}_{x \sim \mathcal{D}} \left[ x \notin [x_1, x_m] \right] \leq \mathbb{P}_{x \sim \mathcal{D}} \left[ x \notin (a, b) \right] = 2\epsilon' = \varepsilon/2$$

**Claim**: Split $[a, b]$ to $\frac{b-a}{\delta}$ intervals of equal size of $\delta$ and denote the intervals by $A_i = [a + i\delta, a + (i+1)\delta)$. Then, letting $m \geq \frac{6(b-a)}{\varepsilon \delta} \log(6(b-a)/\varepsilon\delta)$ where $\delta = \varepsilon/12\lambda$.

$$\mathbb{P}[\exists A_i \text{ s.t. } x \in A_i \text{ and } \forall x_j \in S, x_j \notin A_i] \leq \varepsilon/3.$$

**Proof**: Denote the above event by $B$. Now, let $p_i = \mathbb{P}[x \in A_i]$, by the union bound:

$$
\begin{aligned}
\mathbb{P}(B) &\leq \sum \mathbb{P}[x \in A_i, \forall x_j, x_j \notin A_i] \\
&= \sum_i p_i (1 - p_i)^m \\
&\leq \sum_i p_i e^{-p_i m} \\
&= \sum_{i: p_i < \frac{\delta}{b-a}\varepsilon/6} p_i e^{-p_i m} + \sum_{i: p_i \geq \frac{\delta}{b-a}\varepsilon/6} p_i e^{-p_i m} \\
&\leq \varepsilon/6 + \sum_{i: p_i \geq \frac{\delta}{b-a}\varepsilon/6} e^{-p_i m}
\end{aligned}
$$

Now, since we chose $m \geq \frac{6(b-a)}{\varepsilon\delta} \log(6(b-a)/\varepsilon\delta)$ we have that,

$$P(B) \leq \varepsilon/3.$$

**Claim**: When $m$ defined as above, we have that,

$$\mathbb{E}_x \left[ \sum_y \left| \mathbb{P}_{S \sim \mathcal{D}^m} [\mathcal{A}(S)(x)|x] - \mathbb{P}[y|x] \right| \, \middle| \, x \in [a, b] \right] \leq \varepsilon/2$$

*Proof.* Let $C$ be the event $x \in [a, b]$ intersected with $B^c = \Omega \setminus B$. Then, denote by $x_i$ the nearest neighbor of $x$ in $S$ and assume WLOG $x \in [x_i, x_i + 1]$. Conditioned on $C$, $x - x_i \leq \delta$, thus,

$$
\begin{aligned}
\left| h_{\hat{\theta}}(x) - y_i \right| &= \left| \frac{y_{i+1} - y_i}{x_{i+1} - x_i} (x - x_i) \right| \\
&\leq \frac{|y_{i+1} - y_i|}{2} \leq 1
\end{aligned}
$$

And consequently, the sign of $x$ will be $y_i$. Thus, (conditioning on $C$)

$$
\begin{aligned}
\left| \mathbb{P}_{S \sim \mathcal{D}^m}[\mathcal{A}(S)(x)|x] - \mathbb{P}[y|x] \right| &\leq \left| \mathbb{P}_{S \sim \mathcal{D}^m}[\mathcal{A}(S)(x)|x] - \mathbb{P}[y|x_i] \right| + |\mathbb{P}[y|x_i] - \mathbb{P}[y|x]| \\
&\leq \lambda\delta \leq \varepsilon/12
\end{aligned}
$$

We thus can conclude that,

$$\mathbb{E}_x\left[\sum_y \left|\mathop{\mathbb{P}}_{S\sim\mathcal{D}^m}[\mathcal{A}(S)(x)|x] - \mathbb{P}[y|x]\right| \, \Big| \, x\in[a,b]\right]$$

$$\leq \mathbb{P}(B) + \mathbb{E}_x\left[\sum_y \left|\mathop{\mathbb{P}}_{S\sim\mathcal{D}^m}[\mathcal{A}(S)(x)|x] - \mathbb{P}[y|x]\right| \, \Big| \, C\right] = \varepsilon/2$$

Now combining all of the above conclude the proof of the Theorem. $\qquad\square$

$\hfill\square$

*Proof.* of Theorem 20. First, we show that $\mathcal{A} = \text{ERM}_{\mathcal{H}}$ is a teacher when $m = \frac{2k\log(2k/\varepsilon)}{\varepsilon}$. Let $S = \{(\mathbf{x}_i, y_i)\}_{i=1}^m \sim \mathcal{D}^m$ be the sample set. Also, let $B = \{\mathbf{x}\in\mathcal{X} \mid \mathbb{P}(\mathbf{x}_i = \mathbf{x}) \leq \frac{\varepsilon}{2k}\}$ and $G = \mathcal{X}\setminus B$. We first show that for every $\mathbf{x}\in G$ we have $\mathbb{P}[\mathbf{x}\notin S]\leq\frac{\varepsilon}{2k}$.

$$\mathbb{P}[\mathbf{x}\notin S|\mathbf{x}\in G]\leq(1-\varepsilon/2k)^m\leq e^{-\log(2k/\varepsilon)} = \varepsilon/2k$$

Now we can use the union bound to show that,

$$\mathbb{P}[\exists\mathbf{x}_i\in G\setminus S]\leq\varepsilon/2$$

Using the union bound again, we can see that for a new example $\mathbf{x}'$: $\mathbb{P}[\mathbf{x}'\in B]\leq\frac{\varepsilon}{2}$. Thus, $\mathbb{P}[\mathbf{x}'\in B \text{ or } \exists\mathbf{x}_i\in G\setminus S]\leq\varepsilon$. Now, for each $\mathbf{x}\in S\bigcap G$ the label $y(\mathbf{x})$ given by ERM can be seen as a Condorcet Jury voting by the set of $\{y_i|\mathbf{x}_i = \mathbf{x}\}$. We can use Theorem 1 from [4] that shows that Condorcet Jury voting is monotone in the number of votes. Thus, $\mathbb{P}[f_{\mathcal{D}}^*(\mathbf{x}') = f_{\mathcal{A}(\mathcal{D}^m)}^*(\mathbf{x}')] = 1$ using the aforementioned conditioning. Similarly, we have that $\gamma_\varepsilon(\mathcal{A}(\mathcal{D}^m)) \geq \gamma(\mathcal{D})$ (i.e., $\tau = 0$). As we can condition as before and the CJT monotonicity Theorem implies that the margin can only increase (as the probability of the top label increases). $\qquad\square$

**Lemma 24.** *Let $y_1,\ldots,y_n$ be some independent random variables with $y_i\in\{\pm 1\}$ s.t. $\mathbb{P}(y_i = 1) = p_i$, where either $p_1,\ldots,p_n\in(1/2,1]$ or $p_1,\ldots,p_n\in[0,1/2)$, and let $\gamma = \min_i|2p_i - 1|$. Denote $y^* = \text{sign}(\sum_{i=1}^n y_i)$, and let $\ell(\mathbf{y}) = 2\cdot\sum_{i=1}^n\mathbf{1}\{y_i\neq y^*\}$ and $\tilde\ell(\mathbf{y},r) = n(1-r)$ for some $0 < r\leq\frac{\gamma}{3}$. Then, there exists some universal constant $c > 0$, s.t. for every $\delta\in(0,1)$, if $n\geq\frac{8\log(1/\delta)}{\gamma^2}$ w.p. at least $1-\delta$ we have $\ell(\mathbf{y}) < \tilde\ell(\mathbf{y})$.*

*Proof.* Let $S = \sum_{i=1}^n y_i$. Observe that:

$$\ell(\mathbf{y}) = 2\sum_{i=1}^n\mathbf{1}\{y_i\neq y^*\} = 2\cdot\sum_{i=1}^n\left(\frac{1}{2} - \frac{y_iy^*}{2}\right) = n - y^*\sum_{i=1}^n y_i = n - |S|$$

Also note that $\mathbb{E}[S] = \sum_{i=1}^n(2p_i - 1)$ so $|\mathbb{E}[S]|\geq n\gamma$. Now, from Hoeffding's inequality:

$$\mathbb{P}\left(|S - \mathbb{E}[S]|\geq\frac{n\gamma}{2}\right)\leq 2\exp\left(-n\gamma^2/8\right)\leq\delta$$

So, w.p. at least $1-\delta$ we have:

$$\ell(\mathbf{y}) = n - |S|\leq n - |\mathbb{E}[S]| + |S - \mathbb{E}[S]|\leq n - \frac{n\gamma}{2} < n - rn = \tilde\ell(\mathbf{y})$$

where we use the fact that $r\leq\frac{\gamma}{3} < \frac{\gamma}{2}$. $\qquad\square$

*Proof.* of Theorem 22.

**Claim.** The Bayes optimal classifier $f^*$ on $\mathcal{D}$ is constant on each ball.

**Proof.** Let $\mathbf{x} \in B(\mathbf{c}_i, r)$ and let $y_1 =: f^*(\mathbf{c}_i)$ be the $\arg\max \mathbb{P}_{\mathcal{D}}\left[y|\mathbf{c}_i\right]$. Using the margin condition on $\mathbf{c}_i$ we know that $\mathbb{P}(y_1|\mathbf{c}_i) > \mathbb{P}(y_2|\mathbf{c}_i) + \gamma$ (here $y_2 = -y_1$). Since $\mathbf{x} \in \mathcal{B}(\mathbf{c}_i)$ we know that $d(\mathbf{x}, \mathbf{c}_i) < r$ and using the $\lambda$-Lipschitzness of the distribution we get that,

$$\mathbb{P}(y_1|\mathbf{x}) \geq \mathbb{P}(y_1|\mathbf{c}_i) - \lambda r > \mathbb{P}(y_2|\mathbf{c}_i) + \gamma - \lambda r \geq \mathbb{P}(y_2|\mathbf{x}) + \gamma - 2\lambda r \geq \mathbb{P}(y_2|\mathbf{x})$$

So $f^*(\mathbf{x}) = f^*(\mathbf{c}_i)$ thus $f^*$ on $B(\mathbf{c}_i, r)$ is determined by $f^*(\mathbf{c}_i)$ and therefore constant on the ball. In a similar fashion, we proceed to show that with high probability a hypothesis output by $\mathrm{ERM}_{\mathcal{H}}^{hinge}$ is constant on each ball with significant probability mass.

**Claim.** Let $h \in \mathcal{H}$ be some function that is not constant on $B(\mathbf{c}_i, r)$. Then $\left|\hat{h}(\mathbf{x})\right| \leq 2Lr$ for every $\mathbf{x} \in B(\mathbf{c}_i, r)$.

**Proof.** Fix $\mathbf{x} \in B(\mathbf{c}_i, r)$ and let $\mathbf{x}' \in B(\mathbf{c}_i, r)$ s.t. $\mathrm{sign}\,\hat{h}(\mathbf{x}) \neq \mathrm{sign}\,\hat{h}(\mathbf{x}')$. Observe that $\left|\hat{h}(\mathbf{x}) - \hat{h}(\mathbf{x}')\right| \leq L\|\mathbf{x} - \mathbf{x}'\| \leq 2Lr$. So, if $\hat{h}(\mathbf{x}) > 0$ we get that

$$\hat{h}(\mathbf{x}) \leq \hat{h}(\mathbf{x}) - \hat{h}(\mathbf{x}') \leq 2Lr$$

otherwise if $\hat{h}(\mathbf{x}) \leq 0$ we get that

$$-\hat{h}(\mathbf{x}) \leq \hat{h}(\mathbf{x}') - \hat{h}(\mathbf{x}) \leq 2Lr$$

**Claim.** For each ball $B(\mathbf{c}_i, r)$ with $\mathbb{P}[\mathbf{x} \in B(\mathbf{c}_i, r)] \geq \varepsilon/2k$, if $m \geq \frac{16k\log(2k/\varepsilon)}{\gamma^2\varepsilon}$ we have w.p. at least $1 - \varepsilon/2k$ that $|S \cap B(\mathbf{c}_i, r)| \geq n$ where $n = \frac{8\log(2k/\varepsilon)}{\gamma^2}$.

**Proof.** Let $S = \{(\mathbf{x}_i, y_i)\}_{i=1}^m$ and denote $\xi_i = \mathbf{1}\{\mathbf{x}_i \in B(\mathbf{c}_i, r)\}$, and notice that $|S \cap B(\mathbf{c}_i, r)| = \sum_{i=1}^m \xi_i$. It holds that: $\mathbb{E}\left[\sum_{i=1}^m \xi_i\right] \geq \frac{m\varepsilon}{2k}$. Note, similar to the argument in Theorem 20, if $m \geq \frac{2k\log(1/\delta)}{\varepsilon}$ w.p. $1 - \delta$ it holds that $\sum_{i=1}^m \xi_i \geq 1$. When $m \geq \frac{16k\log(2k/\varepsilon)\log(16k\log(2k/\varepsilon)/\varepsilon\gamma^2)}{\varepsilon\gamma^2}$ we can apply the same argument for each "block" of size $\frac{2k\log(16k\log(2k/\varepsilon)/\varepsilon\gamma^2)}{\varepsilon}$. That is, we are using $\delta = \frac{\varepsilon\gamma^2}{16k\log(\frac{2k}{\varepsilon})}$ and the number of blocks is $n = \frac{8\log(2k/\varepsilon)}{\gamma^2}$ to get that with probability $1 - \delta n = 1 - \frac{\varepsilon}{2k}$ it holds that $\sum_{i=1}^m \xi_i \geq \frac{8\log(2k/\varepsilon)}{\gamma^2}$.

**Claim.** For each ball $B(\mathbf{c}_i, r)$ with $\mathbb{P}[\mathbf{x} \in B(\mathbf{c}_i, r)] \geq \varepsilon/2k$ the probability that $\mathrm{ERM}_{\mathcal{H}}^{hinge}$ is constant on $B(\mathbf{c}_i, r)$ is at least $1 - \varepsilon/2k$.

**Proof.** From the previous two claims it holds that $f^*$ is constant on $B(\mathbf{c}_i, r)$ and that with probability $\geq 1 - \varepsilon/2k$ it holds that $|S \cap B(\mathbf{c}_i, r)| \geq 8\log(2k/\varepsilon)/\gamma^2$. Let $n = |S \cap B(\mathbf{c}_i, r)|$, and denote $(\mathbf{x}_1, y_1), \ldots, (\mathbf{x}_n, y_n)$ the examples in $S \cap B(\mathbf{c}_i, r)$. By definition of $\gamma(\mathcal{D})$ it holds that $\mathbb{P}_{\mathcal{D}}[\tilde{y}_i = 1|\mathbf{c}_i] = p_i$ with $|2p_i - 1| \geq \gamma$. Let $y^* = \mathrm{sign}(\sum_{i=1}^n y_i)$ and let $h^* \in \mathcal{H}$ be a hypothesis s.t. $h^*(\mathbf{x}) = y^*$. Let $h \in \mathcal{H}$ be some function that is not constant on $B(\mathbf{c}_i, r)$. Then:

$$\sum_{i=1}^n \ell_{hinge}(h^*(\mathbf{x}_i), y_i) = 2\sum_{i=1}^n \mathbf{1}\{y_i \neq y^*\} = \ell(\mathbf{y})$$

Observe that from the previous claim we have $|h(\mathbf{x}_i)| \leq 2Lr < 1$ and therefore:

$$\sum_{i=1}^n \ell_{hinge}(h(\mathbf{x}_i), y_i) \geq \sum_{i=1}^n 1 - y_i h(\mathbf{x}_i) \geq \sum_{i=1}^n (1 - |h(\mathbf{x}_i)|) \geq n(1 - 2Lr) = \tilde{\ell}(\mathbf{y}, 2Lr)$$

Therefore, if $r \leq \frac{\gamma}{3L}$, w.p. at least $1 - \varepsilon/2$ we have

$$\sum_{i=1}^n \ell_{hinge}(h^*(\mathbf{x}_i), y_i) \leq \sum_{i=1}^n \ell_{hinge}(h(\mathbf{x}_i), y_i)$$

Thus, using the union bound we get that with with probability $> 1 - \varepsilon$, $\mathrm{ERM}_{\mathcal{H}}^{hinge}$ on each ball with probability mass $\geq \frac{\varepsilon}{2k}$ will be constant. Now, since the Bayes is fixed on each ball, the output hypothesis could be seen as Condorcet Jury voting on each ball independently thus proving (same argument as Theorem 20) both condition 1 and 2. $\qquad\square$

# D  Experimental Details

In this section, we elaborate the exact details used in our experiments. In all experiments, we train ResNet-18 [13] with batch size 128 and 0.0005 weight decay. On CIFAR-10 [16] we train for 50 epochs and for CIFAR-5m we train for 1 epoch using cos-annealing learning rate that starts from 0.05 for both datasets. This optimization procedure achieves $\approx 94\%$ accuracy on CIFAR-10 when train on clean data. However, we add 20% fixed label noise. With label noise the model (without early stopping) has $81.3\%$ accuracy on the clean test set. For each experiment in the body we use (at-least) 10 random seeds. So for example, for the 10 random teachers experiment we train 100 teacher models and chose 10 fixed teachers at random for each student seed.