# OpenReview forum: "Knowledge Distillation: Bad Models Can Be Good Role Models"
_NeurIPS.cc/2022/Conference — NeurIPS 2022 Accept_

### Official Review · Reviewer_Yejy · 2022-06-26

**Rating:** 7
**Confidence:** 4
**Soundness:** 3 good
**Presentation:** 3 good
**Contribution:** 3 good

**Summary:**

The paper offers a theoretical framework for studying learning with label noise. It  shows how to  leverage multiple
independent teachers to approximate the Bayes optimal classifier either via ensembling at inference time or via distillation on unlabeled data. Specifically, it shows that ERM algorithm on noisy data produces  samplers of posterior distribution and as such it replicates the noise learned from the training to the test distribution. The paper then defines a teacher for a distribution with noisy labels. It shows that a sampler is a teacher with a sample complexity that depends on the noise. Next, it shows that ensemble of teachers applied at test time cancels noise.  To reduce the complexity in test time, the paper suggest to distill knowledge from the ensemble to a single student by labeling a new unlabeled set from the same domain by a majority vote over teachers' predictions and training the students on the resulting labeled set using ERM. The student is guaranteed to achieve good performance by standard VC bound (although to avoid overparametrized setting, the unlabeled set should be very large). To reduce the training time, they suggest to replace the ensemble with a teacher chosen an random from a fixed pool. They showed that a student trained with a random teacher will guaranteed to show good performance but with even larger sample complexity. The paper also shows that some well known algorithms are teachers under some conditions over the underlying distribution.
The experiments on CIFAR10 demonstrate  that an ensemble of teachers in test time improves over a single teacher.  Knowledge  distillation from ensemble of  teachers or random teacher via labeling of a large set CIFAR_5m  and training a student on it with ERM yields further improvement in test accuracy.


**Questions:**

Figure 1, when discussed early in the paper is not very clear. It becomes clear later, but maybe some amendment to the caption could be done.

**Limitations:**

I didn't find a discussion of limitations.

**Strengths And Weaknesses:**

Strengths: Nice and solid theoretical paper that addresses an important problem. The empirical results support the theoretical findings.
Weaknesses: Labeling a large external set 1) requires a larger unlabeled  data set from the same domain, which is not practical in many real life applications; 2) it significantly increases the training time.

---

> ### Author Response · Authors · 2022-07-31
> **Reply to reviewer Yejy**
>
> We would like to thank the reviewer for championing our paper.
> We will amend figure 1 according to your suggestions. Regarding the labeling of an external set, while it is not practical to obtain such large unlabeled set, it is however, often easier than obtaining such set and annotating it. Distillation does not help with the first part, but at least it reduces the cost of manual labeling.
>
> - Regarding training time, it is unclear if using a large unlabeled set will be worse in terms of training time than using a smaller labeled set but using more epochs. For example, having 100 epochs of cifar10 (50k samples) is equivalent to just one epoch of cifar5m (5m samples).

---

### Official Review · Reviewer_mfji · 2022-07-11

**Rating:** 6
**Confidence:** 4
**Soundness:** 3 good
**Presentation:** 3 good
**Contribution:** 3 good

**Summary:**

This paper supplements the empirical results of Nakkiran and Bansal by providing a theoretical framework for analyzing the conditional sampling behavior of overparameterized networks. The authors show that obtaining a noisy sampler can be more sample efficient than a good classifier, and that an ensemble of samplers suffices to train an accurate student model, even if each sampler is a bad classifier individually. They also show that common learning algorithms generate samplers given sufficient overparameterization.

**Questions:**

Practically speaking, does the introduced theoretical framework have the potential to explain the success of distillation in practice (when there is typically a single sampler)? Are there any practical recommendations (other than using an ensemble of teacher networks) that the theory predicts?

**Limitations:**

The authors stated that they provided the code, data, and instructions needed to reproduce the main experiments and that all of the training details were provided. But, to the best of my knowledge, there is no additional supplementary material containing this information.

**Strengths And Weaknesses:**

## Strengths
* The work and theoretical techniques are novel. The authors do a good job of covering related work in a concise and detailed way.
* The paper is well-written and organized overall.
* The proposed explanation for the success of distillation and the theoretically-derived recommendation for using an ensemble of teachers are unique and interesting contributions. Although the authors don’t show that deep neural networks are samplers, I can imagine future work building on the techniques presented in the paper.
* The paper makes strong theoretical contributions and the techniques seem to be sound
* The results supplement the empirical results of Nakkiran and Bansal and implicitly provide some justification (assuming that overparameterized NNs are samplers) for the success of overparameterized networks in the context of distillation. It seems likely that future work can build on the theoretical techniques presented in this work.


## Weaknesses
* The motivation for the theoretical framework explicitly comes from the claim that overparameterized neural networks are samplers (as stated in the abstract), but this is not formally shown in the paper. This is an understandable limitation (proving properties of NNs is hard), but the wording in Sec. 4 (“completing the picture”) and the emphasis on NNs throughout the paper (including the NN results in Sec. 5) make it seem like it is already proven
* The experimental result on a single small-scale ML scenario is underwhelming relative to the paper’s theoretical contributions. Given the widespread success of ensembles in ML, it is not surprising that using multiple teachers to conduct distillation is beneficial. The improvements in student performance with using additional teacher models can be due to the smoothening of the training noise when multiple teachers are used, for example, and not necessarily be indicative of the presented theory in action. To that end, I am not sure what novelty lies in Fig. 2 even with the proposed algorithm of randomly selecting a teacher, which is the ensemble’s mean output in expectation
* Minor comment, but Fig. 2 should start out with 1 teacher since the y-axis represents the student test accuracy. Without a teacher it’s not clear what the student performance after distillation even means, and it is confusing that the authors plotted the teacher accuracy for that point
* In practice, distillation is done by training a much larger teacher model to train a small student model. More often than not, the teacher model is a good classifier and an ensemble of networks is not required. I am not sure if or how the theory applies here

---

> ### Author Response · Authors · 2022-07-31
> **Reply to Reviewer mfji**
>
> We would like to thanks the reviewer for supporting our paper.
>
> - We agree that giving a complete proof that overparameterized neural networks are samplers in a more general setting is a very important goal. However, as the reviewer noted, showing such property in full generality in NNs is hard to achieve. That said, we show theoretical results in some limited setting (Theorem 15) as well as empirical results that corroborate the main hypothesis of the paper. We would rephrase our claims to better reflect the formal results shown in the paper.
> - The main emphasis of our paper is on theoretical contribution. While there are many empirical works on knowledge distillation, the field lacks theoretical understanding of why and when knowledge distillation is the best practice. Thus, in our work, we concentrated on broadening our theoretical perspective by giving a concrete framework where knowledge distillation works well. Note that our theory includes an analysis of a novel method of randomly choosing a teacher to label each example, which can potentially reduce the cost of labeling examples relative to averaging over multiple teachers.
> - We will amend figure 2. Thanks for the suggestion.
> - Knowledge distillation is applied in various settings, and clearly our results do not hold in all of them. However, we believe our theory captures a wide variety of settings that are applicable in practice.
> - Regarding the question: “does the introduced theoretical framework have the potential to explain the success of distillation in practice”. Our theoretical analysis assumes multiple teachers trained on different subsets of the data, because we need a source of “randomness” that originates from the random sampling of the data. When only one sampler is used, we believe that other sources of randomness can explain the success of distillation, for example when random augmentations are introduced to the training procedure. This is a very interesting question for a follow-up work.
> - Regarding the question: “Are there any practical recommendations (other than using an ensemble of teacher networks) that the theory predicts”. Our theory suggests that, assuming the data has some amount of noisy labels, choosing a random teacher from an ensemble to label each example is equivalent to labeling using the full ensemble, while reducing the cost of labeling each example significantly. This is indeed corroborated by our experiments.
> - Regarding code for reproducibility, we will upload it upon submission, thanks for taking notice.

---

### Official Review · Reviewer_5UHV · 2022-07-14

**Rating:** 6
**Confidence:** 3
**Soundness:** 3 good
**Presentation:** 3 good
**Contribution:** 2 fair

**Summary:**

The paper establishes the theoretical framework for studying samplers that are defined as a learning algorithm creating models whose aggregated prediction distribution is close to the Bayes optimal via repeated training with independent samples. The authors display that the learner generate "bad models" when inspecting the individual models, but those bad models are data efficiently and jointly can be a good ensemble teacher giving the nearly Bayes labels for distilling a student. The paper additionally provides theoretical guarantees for the samplers and the teachers. From the practical perspective, the authors propose two methods to distill from such a sampler ensemble, and have demonstrated empirical that the distilled students achieve high performance on CIFAR-10.

**Questions:**

The reviewer does not have major concerns regarding the theoretical development in the paper. Here are some minor questions:

1. The title and the abstract of the submission are currently misleading, as the analyses are mostly done to show the ensemble of bad models (samplers) can be a good role (ensemble) model. This on the other hand has slightly weakened the novelty of the submission, as it is a known property of an ensemble to take advantage of its weak learners in general settings.

2. The theoretical analyses here assume that the sampler algorithm A has access to a large number of independent samples to train models  whose joint predicted probability is nearly Bayes optimal. This is somehow contradictory to the fact that "bad models" are usually a consequence of data scarcity, and that independent samples are usually hard to acquire even for ensembles which usually rely on bagging. If so the Bayes optimal would be respect to the empirical distribution, and there is another term to bridge it with the true distribution.

3. Following 2, the empirical study currently reuse same training dataset for getting the ensemble teacher. It would be nice to see how a theoretically perfect setup works in practice.

4. There lacks practical guidance on how to properly choose the hyperpemeters, e.g. m, k in the algorithm. These two values are admittedly related to the sample distribution, so that it would be nice to at least have some estimators.

5. Although easy to follow, the submission appears to be too dense when fitting into 9 pages.


**Limitations:**

Yes. The authors have mentioned both the theoretical limitation of finding hypotheses spaces that meet the requirements of teachers, as well as some technical aspects, e.g. reusing the training datasets for the empirical study.

**Strengths And Weaknesses:**

Strengths:
The paper presents clear and self contained analyses between the learners, samplers and teachers as also defined in the paper. The research question of understanding data inefficient teachers is well motivated and is very meaningful. The theoretical results are concise and sound.

Weakness:
The empirical side of the work is relatively weak, as the practical implementation of the proposed method is costly, and the empirical study in the paper is limited.

---

> ### Author Response · Authors · 2022-07-31
> **Reply to reviewer 5UHV**
>
> We would like to thank the reviewer for supporting our paper. While there are many empirical works on knowledge distillation, the field lacks theoretical understanding of why and when knowledge distillation is a good practice. Thus, in our work, we concentrated on broadening the theoretical perspective by giving a concrete framework where knowledge distillation works well. Note that while our paper proves that ensembles reduce the prediction noise of individual models, a student taught by an ensemble of teachers has similar virtues. This has the benefit of cheap inference and also can be extended (see theorem 12) to the random teacher algorithm, which greatly reduces the cost of labeling each example.
> Answers to some specific questions raised by the reviewer:
>
> - Following your suggestion, we would amend the abstract to better present the novelty of our contributions.
> - In our theoretical setting, we analyze teachers trained on disjoint subsets of the data, as it simplifies our results. However, our experiments show that even when the teachers are trained on the same dataset (with different initial conditions), the benefit of knowledge distillation is observed.

---

> > ### Comment · Reviewer_5UHV · 2022-08-09
> > **response to rebuttal**
> >
> > I would like to thank the authors for the rebuttal addressing my concerns. After reading I would like to keep my initial rating.

---

### Official Review · Reviewer_ANZb · 2022-07-18

**Rating:** 6
**Confidence:** 3
**Soundness:** 3 good
**Presentation:** 2 fair
**Contribution:** 2 fair

**Summary:**

This paper builds upon the observation by Nakkiran and Bansal, 2020 and proposes a theoretical framework to study conditional samplers, which are interpreted as classifiers trained over noisy labeled data. The authors formally define concepts of conditional sampling from a learning perspective. Particularly, the authors relate samplers to knowledge distillation and show that weak samplers can be used as teachers for distillation in an ensemble manner. The authors also propose an algorithm for distillation using labels generated by a random teacher from a fixed pool of samplers. Finally, they show some results for classical classification algorithms.

**Questions:**

1) Is the claim “neural networks trained on noisy data replicate the noise to unseen samples as well” in line 143 – 144 actually proved in the paper? If not, to what extent does this statement hold in general?

2) The first condition in Definition 7 sounds very strong. I am not sure why a sampler that satisfies this condition would not be a learner, if not a “good” learner”?

3) Can you say something about the two sample complexity bounds in Theorem 8 and 9, e.g., how do you compare them? What do you mean by “interpolation between a sampler and a learner”.

4) From Figure 1, it is somewhat unintuitive why the ensemble and random distilled would work: each teacher is a sampler that replicates the noise to unseen data, so I find it difficult to understand why voting would result in a different and correct label to previously incorrectly classified sample.

5) The experiments are conducted on CIFAR-10 with pseudo-random label noises. Please report similar metrics on the original CIFAR-10?


**Limitations:**

See the above.

**Strengths And Weaknesses:**


Strengths:
+ A new theoretical formulation of learning as sampling and the connections between samplers and classifiers.
+ Interesting implications of the formulation to knowledge distillation.
+ A new distillation algorithm.

Weakness:
- The framework relies on an observation that might not always hold, and if so, it is not clear if one needs to distinguish between a sampler and learner in practice, so I don’t see the practicality of the proposed theory framework.
- Lack of validation on real data (to support theory and the proposed algorithm.)

---

> ### Author Response · Authors · 2022-07-31
> **Reply to reviewer ANZb**
>
> We would like to thanks the reviewer for supporting our paper. We address specific concerns raised by the reviewer below:
>
> - We agree that Distributional Generalization (as proposed by Nakkiran and Bansal) is not always guaranteed to hold, for example when the model is under-parameterized noise memorization is limited. However, in many other settings, as observed by Nakkiran and Bansal and corroborated by our experiments, over-parameterized DNNs “replicate” the train noise to the test distribution.
> - “neural networks trained on noisy data replicate the noise to unseen samples as well” - We use this statement as the premise of our paper. This is not fully proved for general settings of training DNNs, nevertheless, it was empirically observed (as mentioned above) and we have theoretical results showing that it provably holds in simple settings (e.g., DNNs trained on scalar data, see Theorem 15).
> - “first condition in Definition 7 sounds very strong”. The first condition means that the predictions of the teacher are biased toward the right label. Note, however, that the teacher can be very noisy. For example, given an image of a cat, if a model returns cat with probability 51% and dog with probability 49% then we say it is a teacher. However, this model is not a good learner because its error is very high.
> - Compare the bounds in Thms 8,9: The bounds in theorems 8 and 9 are in some sense incomparable since they depend on the sample complexities of samplers and learners which can be very different (see theorem 6). What we mean by “interpolation” is that the concept of teachers extends both the definition of sampler and learner, and also captures models that are “in between” samplers and learners.
> - If the noise is not correlated, different teachers can get different examples right. In that case, using an ensemble reduces the variance of the noise.
> - We will report similar metrics for the clean CIFAR-10 data in the final version of the paper.

---

### Meta-Review · Area_Chair_3Qev · 2022-08-24

**Recommendation:** Accept
**Confidence:** Less certain

**Metareview:**

The paper develops a theoretical framework (in the context of learning theory) for training overparameterized networks with label noise, and shows that, in the context of ensemble distillation, teacher networks need not be good classifiers as long as they are good conditional samplers (in the sense defined in the paper).

This is a primarily theoretical paper, with limited empirical results. All reviewers are positive about the paper: one reviewer recommends acceptance (7), the other three recommend weak acceptance (6). The reviewers are positive about the theoretical aspects of the paper, describing them as interesting and technically sound, but are less convinced about the practicality of the theory or the significance of the empirical results.

Given that the theoretical contribution of the paper seems significant and no concerns have been raised about the soundness of the theoretical arguments, I'm happy to recommend acceptance.

**Award:**

No

---

### Decision · Program_Chairs · 2022-09-14

Accept